# Randomized tests for high-dimensional regression: A more efficient and powerful solution

**Yue Li**[1], **Ilmun Kim**[2], **Yuting Wei**[1]*
[1]Department of Statistics and Data Science, Carnegie Mellon University
[2]Department of Pure Mathematics and Mathematical Statistics, University of Cambridge
yuel5@andrew.cmu.edu, ik396@dpmms.cam.ac.uk, ytwei@cmu.edu

## Abstract

We investigate the problem of testing the global null in the high-dimensional regression models when the feature dimension $p$ grows proportionally to the number of observations $n$. Despite a number of prior work studying this problem, whether there exists a test that is model-agnostic, efficient to compute and enjoys a high power, still remains unsettled. In this paper, we answer this question in the affirmative by leveraging the random projection techniques, and propose a testing procedure that blends the classical $F$-test with a random projection step. When combined with a systematic choice of the projection dimension, the proposed procedure is proved to be minimax *optimal* and, meanwhile, reduces the computation and data storage requirements. We illustrate our results in various scenarios when the underlying feature matrix exhibits an intrinsic lower dimensional structure (such as approximate low-rank or has exponential/polynomial eigen-decay), and it turns out that the proposed test achieves *sharp adaptive* rates. Our theoretical findings are further validated by comparisons to other state-of-the-art tests on the synthetic data.

**Keywords:** high-dimensional regression, random projections, $F$-test, minimax optimality, proportional regime

## 1 Introduction

Many applications in modern science and engineering operate in the regime where the number of parameters is comparable to the number of observations. For each dimension, on average, there are only a few samples that are available for statistical inference. This new aspect brings challenges to many traditional tools and methodologies which are often built upon the assumption that the number of observations per dimension goes to infinity.

The current paper concentrates on the high-dimensional linear regression model which is the most commonly used statistical approach to model linear relationships between variables. One set of fundamental problems is to conduct hypothesis testing on the linear parameters, in particular, whether there are signals presented in the observations, and if there are, testing the statistical significance of each feature. Despite a considerably amount of prior work devoted to the study of this problem, whether an optimal testing procedure can be designed for arbitrary feature matrix in the proportional regime ($n$ and $p$ grows proportionally) still remains unsettled. The prior literature that considered this problem often assume the feature dimension is low or has an intrinsic low dimension structure — cases including assuming the parameter being sparse (e.g. [16, 1, 46, 39, 25]), lying in a $L_p$-ellipse or convex cone (e.g. [27, 23, 3, 41, 42]) — or assume the feature matrix is generated from a standard Gaussian design in the proportional regime (see, e.g. [38, 26, 36, 31, 12]).

This paper aims to tackle this fundamental problem in the setting where *(i)* no prior knowledge is assumed on the coefficient vector; *(ii)* the feature dimension and the observation size grow proportionally with each other; *(iii)* the fewest number of assumptions are imposed on the design matrix while being adaptive to the cases when the design matrix enjoys simpler structures. Formally, given $n$ i.i.d. data pairs $(\boldsymbol{x}_1, y_1), (\boldsymbol{x}_2, y_2), \ldots, (\boldsymbol{x}_n, y_n)$, with $\boldsymbol{x}_i \in \mathbb{R}^p$ and $y_i \in \mathbb{R}$, generated from a linear model, we have

$$y_i = \langle \boldsymbol{x}_i, \boldsymbol{\beta} \rangle + \sigma z_i,$$

for some unknown vector $\boldsymbol{\beta} \in \mathbb{R}^p$. Here $\langle \cdot, \cdot \rangle$ denotes the standard inner product. The first step in diagnosis of linear models is to test the joint significance of all covariates, namely, test the hypothesis

$$H_0 : \boldsymbol{\beta} = \mathbf{0} \quad \text{versus} \quad H_1 : \boldsymbol{\beta} \neq \mathbf{0}. \tag{1}$$

The classical and commonly used approach for the above testing problem is based on a global $F$-test [27], which is known to be most powerful when the feature dimension $p$ is held fixed as the number of samples $n$ increases. However, it is also widely known that the $F$-test loses its power as the ratio $p/n$ increases when $p$ is comparable but smaller to $n$ (see e.g. [47]). Moreover, when the dimension exceeds the sample size, the $F$-test is not applicable anymore due to the singularity of the sample covariance matrix. In these scenarios, [47] proposed a test based on a $U$-statistic of fourth order and [15] followed up with a modified test that improves power moderately. However, these proposed tests have low power and can be rather sub-optimal in many scenarios. We therefore ask the following question:

> *Whether there exists an analogue of the classical $F$-test that is both efficient to compute and enjoys an optimal power in the regime where $p/n \in (0, \infty)$?*

In this paper, we answer this question in the affirmative using the random projection techniques. Random projection, also known as "sketching", is based on the idea of storing only a (smartly designed) sketched version of the original data (often of lower dimensions), and performing learning algorithms on the sketched version. It is now a standard technique for reducing data storage and computational costs, and promote privacy. These advantages have motivated a lot of studies on designing efficient and differential-private estimation schemes (e.g. [7, 28, 34, 32]), however, the statistical behaviors of test statistics based on random projections are not as well understood (e.g., [14, 24, 13, 29]). The closest to our work is by [29] where a random-projection based Hotelling's $T$-test is proposed for testing the two-sample mean problem, given independent Gaussian samples.

Having set up the problem, we introduce a novel testing procedure, termed *sketched $F$-test* for the testing problem (1). Intuitively, this test can be viewed as a two-step procedure: first each sample point $\boldsymbol{x}_i$ is projected to a $k$-dimensional random subspace for a preselected dimension $k$ and a random subspace $S_k$; and then a classical $F$-statistic is performed to check if the linear coefficients are zero between the projected data matrix and response vector.

**Evaluations:** To evaluate the efficiency and asymptotic power for our proposed test, we compute its asymptotic power function and compare with the state-of-the-art tests ([47, 15]) in terms of asymptotic relative efficiency (ARE). We demonstrate our sketched $F$-test is computationally efficient and has increased power in various scenarios. In the case when the design matrix has a lower intrinsic dimension, the case where the random projections are mostly powerful for, we show that it is sufficient to choose the sketched dimension proportional to the statistical dimension of the design matrix (defined in the sequel), without any loss of statistical accuracy. More precisely, the sketched $F$-test is proved to be asymptotically most powerful. We also demonstrate the advantages of using our proposed test over its state-of-the-art competitors with higher asymptotic power and improved performance on synthetic data.

## 1.1 Our contributions

The main contributions of this paper are summarized below, all of which are built upon a careful analysis of a sketched version of the classical $F$-test.

- In Section 2, we introduce a sketched $F$-test which does not restrain $n, p$ as in prior literature. The explicitly characterizations of its asymptotic power function are provided in Theorem 1. Compared to prior work on this problem, our theoretical results are model-agnostic — generalizing to scenarios that are other than Gaussian design and Gaussian noise.

- We provide a systematic way of selecting the projection dimension based on the intrinsic dimension of the population design matrix $\boldsymbol{\Sigma}$, which is current lacking in the state-of-art testing procedures concerning random projections. When $\boldsymbol{\Sigma}$ is indeed of lower dimensions, which is the case underlying most applications, our proposed test yields adaptive testing rates that are minimax optimal, and fully preserves the signal in the original model. These results are summarized in our Theorem 2 and Theorem 3.

- Compared with the state-of-the-art tests, we demonstrate our superiority in various scenarios both theoretically and empirically, in terms of computational efficiency and asymptotic relative efficiency. We refer the readers to Section 2.3 for theoretical justifications and Section 4 for experimental studies respectively.

## 1.2 Other related work

Related to the global testing problem, there has been an intensive line of work studying procedures that identify non-zero coefficients in high dimensional regression models. To provide theoretical justifications, these procedures often pose more stringent assumptions on the model itself, such as sparsity (e.g. [46, 25, 11]), independence or positive dependence between $p$-values (e.g. [6, 19]). In addition, the resulting theoretical guarantees mainly focus on the type-I error control, without a characterization of the statistical power (e.g. [4, 9, 21]). As a matter of fact, the global testing problem considered in this paper is often regarded as a first stage analysis in practice (see [44, 30]) and is intrinsically easier than testing individual coefficients. Therefore it allows us to derive a refined analysis of its power under much weaker assumptions.

**Notation.** We use $\overset{d}{=}$ for two random variables that have the same distribution. Let $\Phi(\cdot)$ denote the CDF of $\mathcal{N}(0, 1)$, and $z_\alpha$ denote the upper $\alpha$ quantile of $\mathcal{N}(0, 1)$. The upper $\alpha$ quantile of $F$-distribution with degrees of freedom $(p, n - p)$ is denoted by $q_{\alpha, p, n-p}$. Moreover, the norm $\| \cdot \|_2$ stands for Euclidean norm for a vector, and spectral norm for a matrix. Matrix Frobenuis norm is denoted by $\| \cdot \|_F$. We call $a_n \asymp b_n$ if there is a universal constant $c_0$ such that $\frac{1}{c_0} \leq \frac{a_n}{b_n} \leq c_0$ for large enough $n$, and $a_n \lesssim b_n$ if $a_n \leq c_0 b_n$ for large enough $n$.

## 2 Sketched $F$-test

In this section, we formally introduce the proposed sketched $F$-test and describe our main theoretical results along with some necessary background. We start our discussion by reviewing the classical $F$-test and describe why it fails in the high-dimensional setting.

### 2.1 Classical $F$-test

We find it useful to first formulate the observation model in the matrix form. Let $\boldsymbol{y} = (y_1, \ldots, y_n)^\top$ and $\boldsymbol{X} \in \mathbb{R}^{n \times p}$ be the matrix with rows $\boldsymbol{x}_1^\top, \ldots, \boldsymbol{x}_n^\top$, and we can write

$$\boldsymbol{y} = \boldsymbol{X}\boldsymbol{\beta} + \sigma\boldsymbol{z}. \tag{2}$$

Given i.i.d. samples $\{\boldsymbol{x}_i, y_i\}_{i=1}^n$ from model (2) with $n > p$, the classical $F$-test statistic is defined as

$$F = \frac{\widehat{\boldsymbol{\beta}}^\top (\boldsymbol{X}^\top \boldsymbol{X}) \widehat{\boldsymbol{\beta}} / p}{\widehat{\sigma}^2},$$

where $\widehat{\boldsymbol{\beta}} := (\boldsymbol{X}^\top \boldsymbol{X})^{-1} \boldsymbol{X}^\top \boldsymbol{y}$ is the least square estimator and $\widehat{\sigma}^2 := \|\boldsymbol{y} - \boldsymbol{X}\widehat{\boldsymbol{\beta}}\|_2^2 / (n - p)$ is an unbiased estimator of $\sigma^2$. Under the null hypothesis $H_0$, it is well-known that the $F$-test statistic follows the $F$-distribution with $(p, n - p)$ degrees of freedom, whereas under the alternative $H_1$, it follows a noncentral $F$-distribution with $(p, n - p)$ degrees of freedom with the noncentrality parameter $\boldsymbol{\beta}^\top (\boldsymbol{X}^\top \boldsymbol{X}) \boldsymbol{\beta} / 2\sigma^2$. In this setup, the $F$-test rejects the null hypothesis if $F \geq q_{\alpha, p, n-p}$ and its theoretical properties have been well-established in classical settings [see, 33].

As in [47] (and also in [2, 8, 5, 17]), we consider a tractable model where the design matrix $\boldsymbol{X}$ is randomly generated: each row of the design matrix $\boldsymbol{x}_i^\top \in \mathbb{R}^p$ is independently drawn from a multivariate distribution with covariance $\boldsymbol{\Sigma}$. We start by assuming the design matrix follows a multivariate Gaussian distribution with general covariance $\boldsymbol{\Sigma}$ and then generalize some of our results to incorporate other random designs. Random designs enable us to carry out our analysis in a refined manner with tools from random matrix theory and large deviation theory.

Under this random design framework, it is easily seen that the power of the $F$-test is determined by the signal strength $\boldsymbol{\beta}^\top \boldsymbol{\Sigma} \boldsymbol{\beta}$, which is proportional to the expected value of the noncentrality parameter. When this signal strength is of constant order, the testing problem becomes trivial in a sense that the null and alternative hypotheses can be easily distinguished in the limit. This motivates us and others to consider the *local alternative* in which $\boldsymbol{\beta}^\top \boldsymbol{\Sigma} \boldsymbol{\beta}$ diminishes as the sample size goes to infinity. Such framework is standard in asymptotic theory [see, e.g. 40] and has been considered by [47, 35, 15] among others. Under this local alternative, the following lemma studies the asymptotic power of the $F$-test in the regime where $p/n \to \delta \in (0, 1)$. This result is the key ingredient to Theorem 1 in which we study the asymptotic power of the sketched version of the $F$-test (see also [35]).

**Lemma 1.** *Suppose the design matrix $\boldsymbol{X}$ is generated from a multivariate Gaussian distribution with covariance matrix $\boldsymbol{\Sigma}$, and the noise vector $\boldsymbol{z} \sim \mathcal{N}(\boldsymbol{0}, \boldsymbol{I}_n)$. Suppose the signal strength $\boldsymbol{\beta}^\top \boldsymbol{\Sigma} \boldsymbol{\beta} = o(1)$ and $\delta_n = p/n \to \delta \in (0, 1)$ as $n \to \infty$, then the power of the classical $F$-test (defined as $\Psi_n^F := P(F \geq q_{\alpha,p,n-p})$), satisfies*

$$\Psi_n^F - \Phi\left(-z_\alpha + \sqrt{\frac{(1-\delta)n}{2\delta}} \frac{\boldsymbol{\beta}^\top \boldsymbol{\Sigma} \boldsymbol{\beta}}{\sigma^2}\right) \to 0.$$

In our Appendix C.2, we give a proof of this result which significantly simplifies the proof of Theorem 2.1 in [35] and serves as a building block for our other results.

## 2.2 The sketched $F$-test

It is clear from Lemma 1 that the classical $F$-test is not applicable when $p > n$ and performs poorly when $p/n$ is close to one. The main issue arises from the high variance of estimating $\boldsymbol{\Sigma}^{-1}$ by inverting the sample covariance matrix. In this section, we tackle this problem by leveraging the random projections or sketching techniques. The usual purpose of sketching is to conduct dimension reduction while preserving the sample pairwise distances, however, this technique is employed here to reduce the variance in estimating the population covariance matrix. We witness a significant gain in testing power in various high-dimensional scenarios.

Our testing procedure is summarized in the Algorithm 1 as follows.

---

**Algorithm 1** Sketched $F$-test

---

**Input:** data matrix $\boldsymbol{X} \in \mathbb{R}^{n \times p}$, response vector $\boldsymbol{y} \in \mathbb{R}^n$, a sketching dimension $k < n$
**Output:** testing result for linear model (1).
**Step 1:** generate a sketching matrix $S_k \in \mathbb{R}^{p \times k}$ with i.i.d. $\mathcal{N}(0, 1)$ entries;
**Step 2:** compute the least square regression estimate $\widehat{\boldsymbol{\beta}}^S := (S_k^T \boldsymbol{X}^\top \boldsymbol{X} S_k)^{-1} S_k^\top \boldsymbol{X}^\top \boldsymbol{y}$;
**Step 3:** calculate the sketched $F$-test statistic

$$F(S_k) := \frac{\boldsymbol{y}^\top \boldsymbol{X} S_k \widehat{\boldsymbol{\beta}}^S / k}{\|\boldsymbol{y} - \boldsymbol{X} S_k \widehat{\boldsymbol{\beta}}^S\|_2^2 / (n-k)}; \tag{3}$$

**Step 4:** if $F(S_k) \geq q_{\alpha,k,n-k}$, reject $H_0$; otherwise accept $H_0$.

---

A few remarks are in order. First throughout this paper, the projection dimension is selected to be $k < \min\{\mathrm{rank}(\boldsymbol{X}), \mathrm{rank}(\boldsymbol{\Sigma})\}$. With this choice, when $S_k$ has i.i.d. $\mathcal{N}(0, 1)$ entries, $S_k^\top \boldsymbol{X}^\top \boldsymbol{X} S_k$ is invertible almost surely. We make this fact clear in Appendix C.3, which guarantees Algorithm 1 is well-defined even when $p$ is much larger than $n$. Also note that under $H_0$, for any given $S_k$ and any realization $\boldsymbol{X}$, the sketched $F$-test statistic (3) follows the $F$-distribution with $(k, n-k)$ degrees of freedom, and thus the proposed test is a valid level $\alpha$ test. We call $S_k$ from Algorithm 1 a Gaussian sketching matrix with sketching dimension $k$.

In the following, we define a quantity that plays a key role in our further development, namely

$$\Delta_k^2 := \boldsymbol{\beta}^\top \boldsymbol{\Sigma} S_k (S_k^\top \boldsymbol{\Sigma} S_k)^{-1} S_k^\top \boldsymbol{\Sigma} \boldsymbol{\beta}. \tag{4}$$

It can be shown that $\Delta_k$ indeed determines the asymptotic power of the sketched $F$-test. As a consequence of Lemma 1 applying to the sketched dataset, we establish the following guarantee on the asymptotic power of the sketched $F$-test.

**Theorem 1.** *Suppose the design matrix $\boldsymbol{X}$ is generated from a multivariate Gaussian distribution with covariance matrix $\boldsymbol{\Sigma}$, and the noise vector $\boldsymbol{z} \sim \mathcal{N}(\boldsymbol{0}, \boldsymbol{I}_n)$. Assume $\boldsymbol{\beta}^\top \boldsymbol{\Sigma} \boldsymbol{\beta} = o(1)$ and*

$\rho_n = k/n \to \rho \in (0,1)$ *as* $n \to \infty$. *Then, for almost all sequences of sketching matrix* $S_k$, *the power function of the sketched F-test, that is* $\Psi_n^S(S_k) = P(F(S_k) > q_{\alpha,k,n-k})$, *satisfies*

$$\Psi_n^S(S_k) - \Phi\left(-z_\alpha + \sqrt{\frac{(1-\rho)n}{2\rho}} \frac{\Delta_k^2}{\sigma^2}\right) \to 0. \tag{5}$$

We emphasize that the approximation of $\Psi_n^S(S_k)$ to the normal distribution function is precise in the limit including *all constant factors*. This asymptotic expression allows us to compare the proposed test with existing competitors in terms of the asymptotic relative efficiency in Section 2.3. It is also helpful to notice that conditional on $S_k$, the sketched $F$-test can be simply viewed as the original $F$-test applied to the projected data set $XS_k$. The proof is provided in Appendix C.1.

## 2.3 Testing power comparisons

With the asymptotic power function characterized in expression (5), we compare our sketched $F$-test with other existing tests and highlight the advantages of our projection-based approach. First we make note that [47] introduced a test based on a fourth-order $U$-statistic and considered the local alternative $\boldsymbol{\beta}^\top \boldsymbol{\Sigma} \boldsymbol{\beta} = o(1)$ as well as other regularity conditions including

$$\mathrm{tr}(\boldsymbol{\Sigma}^4) = o(\mathrm{tr}^2\{\boldsymbol{\Sigma}^2\}). \tag{6}$$

We refer to their test as ZC test for simplicity. Under this asymptotic setting, the authors showed that the power of ZC test, denoted by $\Psi_n^{ZC}$, satisfies

$$\Psi_n^{ZC} := \Phi\left(-z_\alpha + \frac{n\|\boldsymbol{\Sigma}\boldsymbol{\beta}\|_2^2}{\sigma^2\sqrt{2\mathrm{tr}(\boldsymbol{\Sigma}^2)}}\right). \tag{7}$$

As a follow-up, [15] proposed another $U$-statistic that improves the computational complexity of [47], while achieving the same local asymptotic power specified in expression (7). Given these explicit power characterizations (5) and (7), the rest of this section is dedicated to comparing these methods in terms of a classical measure: the so called "Asymptotic Relative Efficiency", which is explained as follows.

**Asymptotic Relative Efficiency (ARE):** In asymptotic statistics literature, a common way for comparing performances between different testing procedures is based on their asymptotic relative efficiency (ARE) [see, e.g. 40]. Formally, given two level $\alpha$ tests $\phi_1$ and $\phi_2$, the relative efficiency of $\phi_1$ to $\phi_2$ is defined as the ratio $n_2/n_1$ where $n_1$ and $n_2$ are the sample sizes required for $\phi_1$ and $\phi_2$ to achieve the same power against the same alternative. The ARE is then defined as the limiting value of this relative efficiency. Given this definition and building on the power expressions (5) and (7), the ARE of ZC test to our sketched $F$-test is given by the limit of

$$\mathrm{ARE}_n(\Psi_n^{ZC}; \Psi_n^S) := \frac{\sqrt{n}}{\sqrt{\mathrm{tr}(\boldsymbol{\Sigma}^2)}} \bigg/ \sqrt{\frac{1-\rho}{\rho}} \frac{\Delta_k^2}{\|\boldsymbol{\Sigma}\boldsymbol{\beta}\|_2^2}. \tag{8}$$

From the definition, it is clear that

$$\mathrm{ARE}_n(\Psi_n^{ZC}; \Psi_n^S) < 1 \implies \textbf{sketched } F\textbf{-test is preferred}.$$

The rest of this section aims to examine cases where $\mathrm{ARE}_n(\Psi_n^{ZC}; \Psi_n^S) < 1$ with high probability. To facilitate our analysis, we first consider the case when the scaled $\boldsymbol{\beta}$ satisfies the assumption below. Remark that we adopt a frequentist approach throughout the paper, and this assumption is made only to illustrate the performance of the proposed testing procedure in an average sense.

**(A)** *the normalized vector* $\boldsymbol{\Sigma}^{1/2}\boldsymbol{\beta}/\|\boldsymbol{\Sigma}^{1/2}\boldsymbol{\beta}\|_2$ *is uniformly distributed on the* $p$-*dimensional unit sphere, which is independent of* $S_k$.

Intuitively, Assumption **(A)** holds when there is no preferred direction of the alternatives in which the scaled $\boldsymbol{\beta}$ differs from the zero vector. This assumption is standard when there is no prior information available for the alternatives; in particular, [29] imposes a similar assumption in the context of two-sample mean testing. Under Assumption **(A)** and the regularity condition (6), the next proposition provides an upper bound for $\mathrm{ARE}_n(\Psi_n^{ZC}; \Psi_n^S)$ that holds with high probability.

**Proposition 1.** *Under the condition* (6) *and Assumption (A), the following inequality holds with probability* $1 - o(1)$ *as* $n \to \infty$:

$$\mathrm{ARE}_n(\Psi_n^{ZC}; \Psi_n^S) \leq \frac{4}{\sqrt{\rho(1-\rho)}} \frac{\mathrm{tr}(\boldsymbol{\Sigma})}{\sqrt{\mathrm{tr}(\boldsymbol{\Sigma}^2)}} \frac{1}{\sqrt{n}}. \tag{9}$$

We note that the condition (6) is required only for the ZC test, not for our sketched $F$-test. It essentially means that the eigenvalues of $\boldsymbol{\Sigma}$ should not decay too fast. More importantly, if this eigenvalue condition is violated, then ZC test may not be a valid level $\alpha$ test even asymptotically. In such case, it is not meaningful to compare the power of the given tests. In sharp contrast, the power expression (5) for the sketched $F$-test holds regardless of the eigenvalue condition.

**How do we select the dimension of $S_k$?** In Proposition 1, the upper bound of $\mathrm{ARE}_n(\Psi_n^{ZC}; \Psi_n^S)$ is maximized when $\rho = 1/2$. In other words, when there is no extra information on the model, $k = \lfloor n/2 \rfloor$ is a good choice of the sketching dimension to obtain higher asymptotic power. A similar choice was recommended by [29] for the sketched version of Hotelling's $T^2$ test. In Section 3, we explain how to leverage the information of $\boldsymbol{\Sigma}$ and further improve the power.

With the recommended choice of $k = \lfloor n/2 \rfloor$, we make the upper bound (9) more concrete in the following example, whose derivation is postponed to Appendix C.5.

**Example 1.** *If the covariance matrix is well-approximated by a $\sqrt{n}$ dimensional matrix, i.e. for some constant $\epsilon \in (0, 1)$, the first $s = \sqrt{n}$ eigenvalues of $\boldsymbol{\Sigma}$ satisfies*

$$\lambda_1 + \cdots + \lambda_s \geq (1 - \epsilon) \cdot \mathrm{tr}(\boldsymbol{\Sigma}).$$

*Then we have $\mathrm{ARE}_n(\Psi_n^{ZC}; \Psi_n^S) \lesssim 1/n^{1/4}$ with probability $1 - o(1)$.*

## 3    Optimal guarantees for structured designs

In this section, we discuss the case when $(\boldsymbol{\beta}, \boldsymbol{\Sigma})$ are not completely full dimensional, but instead have intrinsically lower dimensional structure, which is commonly observed in applications (such as [18, 37, 10, 20, 22]). We first define our measure of intrinsic dimension $r$ in Section 3.1. We then justify the optimality of choice $k = O(r)$ in two ways: in Section 3.2, we discuss the minimax optimality within the dim-$r$ class; in Section 3.3, we show that this choice fully preserves the signal strength and yields a non-random power expression of the sketched $F$-test. We provide several examples to demonstrate the consequences of the above results.

### 3.1    Intrinsic dimensions

We start by introducing some notation to describe the spectral structure of $\boldsymbol{\Sigma}$. First denote the singular value decomposition of $\boldsymbol{\Sigma}$ as $\boldsymbol{\Sigma} = \boldsymbol{U}\boldsymbol{\Lambda}\boldsymbol{U}^\top$, where the diagonals of $\boldsymbol{\Lambda}$ are aligned in descending order. Write $\boldsymbol{\Lambda} = \mathrm{diag}(\lambda_1, \ldots, \lambda_p)$. Also define a rotated version of our signal as $\widetilde{\boldsymbol{\beta}} = \boldsymbol{U}^\top \boldsymbol{\beta}$. Note that $\boldsymbol{\Lambda}$ and $\widetilde{\boldsymbol{\beta}}$ describe variance and "coefficient" of the model in terms of orthogonalized feature dimensions, and are critical in determining the intrinsic structure of the model. In the following, we provide the definition of intrinsic dimension in a specific case; a more general definition can be found in Appendix B.

**Definition 1** (Informal)**.** *When each $\widetilde{\beta}_i$ follows the same distribution $\mathcal{P}$, we say model (2) has intrinsic dimension up to $r$, if we can find $\eta = o(1)$ and $r \leq p$, such that*

$$\sum_{i=r+1}^p \lambda_i \leq \eta \sum_{i=1}^p \lambda_i \quad \text{and} \quad r\lambda_{r+1} \leq \eta \sum_{i=1}^p \lambda_i. \tag{10}$$

*Denote the collection of such $(\boldsymbol{\beta}, \boldsymbol{\Sigma})$ as $\mathcal{D}(r)$.*

### 3.2    Minimax optimality

In the following, we describe our results under the classical minimax testing framework [e.g. 23] and Gaussian design model (2). Given observations $(\boldsymbol{X}, \boldsymbol{y})$, consider the problem of testing

$$H_0 : \boldsymbol{\beta} = \boldsymbol{0} \quad \text{v.s.} \quad H_1 : (\boldsymbol{\beta}, \boldsymbol{\Sigma}) \in \Theta_r(\tau),$$

in which the alternative space $\Theta_r(\tau)$ is specified by

$$\Theta_r(\tau) = \{(\boldsymbol{\beta}, \boldsymbol{\Sigma}) \in \mathcal{D}(r) : \boldsymbol{\beta}^\top \boldsymbol{\Sigma} \boldsymbol{\beta} \geq \tau^2\}, \quad \text{for some } \tau > 0.$$

Note that the alternative is measured in terms of the Mahalanobis norm instead of the $\ell_2$ norm. We call $\psi$ a level-$\alpha$ test function, if $\psi$ is a measurable mapping from all possible values of $(\boldsymbol{X}, y)$ to

$\{0, 1\}$, and satisfy $\mathbb{E}\left[\psi | H_0\right] \le \alpha$. The Type II error of $\psi$ and the minimax Type II error over $\Theta_r(\tau)$ are defined as

$$r(\psi, \boldsymbol{\beta}, \boldsymbol{\Sigma}) := \mathbb{E}_{\boldsymbol{\beta}, \boldsymbol{\Sigma}} \left[1 - \psi\right]; \quad \mathcal{R}_r(\tau) := \inf_{\psi} \sup_{(\boldsymbol{\beta}, \boldsymbol{\Sigma}) \in \Theta_r(\tau)} r(\psi, \boldsymbol{\beta}, \boldsymbol{\Sigma}),$$

in which the infimum is taken over all level-$\alpha$ test functions. We say level-$\alpha$ test $\psi^*$ is rate optimal with radius $\epsilon_n$, if for any $\gamma \in (\alpha, 1)$, there exists a universal constant $c$, such that the following holds:

*(i) (lower bound)* when $\tau_n \le c\epsilon_n$, $\mathcal{R}_r(\tau_n) \ge \gamma$;

*(ii) (upper bound)* when $\tau_n/\epsilon_n \to \infty$, for any $(\boldsymbol{\beta}_n, \boldsymbol{\Sigma}_n) \in \Theta_r(\tau_n)$, we have $\lim_{n \to \infty} r(\psi^*, \boldsymbol{\beta}_n, \boldsymbol{\Sigma}_n) = 0$.

We are now able to state our optimality result in terms of the testing radius $\epsilon_n$.

**Theorem 2.** *The sketched F-test is minimax rate optimal over $(\boldsymbol{\beta}, \boldsymbol{\Sigma}) \in \mathcal{D}(r)$ with radius*

$$\epsilon_n^2 = \frac{r^{1/2}}{n}, \tag{11}$$

*and the upper bound is reached by choosing sketching dimension $k = O(r)$ and $r \le k$.*

**Remark 1.** The rate above is the same as the global testing rate of linear regression model with feature dimension $r$, sample size $n$, $\boldsymbol{\Sigma} = \mathbf{I}_r$ and $r < n$ [see 11]. In this sense, the intrinsic dimension $r$ measures the minimal number of orthogonal dimensions needed to approximate the original model.

### 3.3 Refined power guarantees

In the following, we provide a more precise characterization of the power function for our proposed test, with the sketching dimension $k$ chosen proportionally to the intrinsic dimension $r$.

**Theorem 3.** *Suppose $(\boldsymbol{\beta}, \boldsymbol{\Sigma}) \in \mathcal{D}(r)$. Assume $\boldsymbol{\beta}^\top \boldsymbol{\Sigma} \boldsymbol{\beta} = o(k/n)$. Then, for almost all sequences of sketching matrix $S_k$, the power function of the sketched F-test satisfies*

$$\Psi_n^S - \Phi\left(-z_\alpha + \frac{\sqrt{n}\boldsymbol{\beta}^\top \boldsymbol{\Sigma} \boldsymbol{\beta}}{\sigma^2} \sqrt{\frac{1 - k/n}{2k/n}}\right) \xrightarrow{p} 0.$$

The proof is provided in the Appendix C.7. Note that in Theorem 3, the quantity inside the second $\Phi$ function is completely deterministic. Now we make a few remarks with regard to the above result.

**Remark 2 (Choice of $k = O(r)$).** Theorem 3 reveals that by choosing $k = O(r)$, we are able to recover the same signal strength as of the original model; increasing the sketching dimension does not increase the power. Together with Theorem 2, we justify the choice of $k = O(r)$.

**Remark 3 (Relaxation of the Gaussian assumption).** So far we have assumed that the design matrix $X$ and random errors $z$ follow Gaussian distributions, mainly to simplify our presentation. Indeed, Theorem 1 and Theorem 3 hold under mild moment conditions on $X$ and $z$. We defer the technical details of this result to the Appendix A.

**Remark 4 (Computational Complexity).** The overall time complexity of the proposed test is $O(npk)$, which can be further reduced by fast Hadamard transform [see, e.g. 45]. In contrast, the $U$-statistics in [47] and [15] have time complexity of $O(n^4 + n^2 p)$ and $O(n^2 p)$, respectively. This illustrates that a computational benefit of the proposed approach over the competitors especially when the design matrix has a low dimensional structure.

**Example 2.** *With $\eta$ set as $1/\log p$ (cf. Definition 1), the intrinsic dimension for several structured designs are computed and summarized in the following table, which have been shown to play a key role in many other problems (e.g. [45, 43]). For more details, see Appendix C.8.*

Table 1: Examples of intrinsic dimension

|  | Coefficient structure | Covariance structure | Intrinsic dimension |
|---|---|---|---|
| $\alpha$-polynomial decay | homogeneous $\widetilde{\beta}_i$ | $\lambda_j \propto j^{-\alpha}$ with $\alpha > 1$ | $r \lesssim (\log p)^{\frac{1}{\alpha - 1}}$ |
| $\gamma$-exponential decay | homogeneous $\widetilde{\beta}_i$ | $\lambda_j \propto \exp(-j^\gamma)$ with $\gamma > 0$ | $r \lesssim (\log \log p)^{\frac{1}{\gamma}}$ |
| structured coefficient | $0 < c_1 \le \widetilde{\beta}_i \sqrt{i} \le c_2$ | $\lambda_j \propto j^{-1}$ | $r \lesssim (\log p)^3$ |

Table 2: Type I and Type II Error Rates with $(n, p) = (50, 500)$. Results are based on 500 repeats.

| | | Slow-decay | | | Fast-decay | | |
|---|---|---|---|---|---|---|---|
| | | $H_0: \|\boldsymbol{\beta}\|_2 = 0$ | $\|\boldsymbol{\beta}\|_2 = 1$ | $\|\boldsymbol{\beta}\|_2 = 5$ | $H_0: \|\boldsymbol{\beta}\|_2 = 0$ | $\|\boldsymbol{\beta}\|_2 = 1$ | $\|\boldsymbol{\beta}\|_2 = 5$ |
| | **Sketching** | 3.2% | **1.4**% | **0.0**% | **4.0**% | **1.4**% | **2.4**% |
| $\|\boldsymbol{\Sigma}\|_F = 100$ | CGZ | 6.0% | 5.4% | 4.6% | 6.2% | 10.4% | 12.4% |
| | ZC | **2.1**% | 16.8% | 0.6% | 4.2% | 14.7% | 6.3% |
| | **Sketching** | 5.4% | **0.0**% | **0.0**% | 4.6% | **1.2**% | **1.6**% |
| $\|\boldsymbol{\Sigma}\|_F = 300$ | CGZ | 5.4% | 4.4% | 3.0% | 5.8% | 12.4% | 12.2% |
| | ZC | **4.2**% | 6.3% | 8.4% | **4.2**% | 8.4% | 10.5% |

## 4 Simulation studies

In this section, we conduct some empirical studies to validate our theoretical findings using synthetic data. We start by comparing the performance of our sketched $F$-test to the existing tests proposed by [47] (referred to as ZC) and [15] (referred to as CGZ) under several scenarios. In the second part, we show that the sketched signal strength $\Delta_k^2$ (cf.(4)) follows closely to that of the original coefficients $\boldsymbol{\beta}^\top \boldsymbol{\Sigma} \boldsymbol{\beta}$. Our experiments are set in the same way as that of [47] and [15], to give a fair comparison to the other two methods.

**Power evaluation.** We first demonstrate the empirical power performance of the tests by varying the decay rate of eigenvalues. For the slow-decay case, we consider $\boldsymbol{\Lambda}$ with $\lambda_i = \log^{-2}(i+1)$ where the projection dimension is chosen to be $k = \lfloor n/2 \rfloor$. Whereas for the fast-decay case, we consider $\lambda_i = i^{-2/3} \log^{-1}(i+1)$ and $k = \lfloor 2 \log p \rfloor$.

Each $\beta_i$ is sampled from $\text{Binomial}(3, 0.3) + 0.3\mathcal{N}(0, 1)$ independently to generate heterogeneous and non-sparse signals; with each given $\boldsymbol{\Lambda}$, let $\boldsymbol{U}$ be column vectors of a QR decomposition of a $p \times p$ matrix with i.i.d. $\mathcal{N}(0, 1)$ entries, and set $\boldsymbol{\Sigma} = \boldsymbol{U}\boldsymbol{\Lambda}\boldsymbol{U}^\top$. We generate $\boldsymbol{Z}$ with i.i.d. $\mathcal{N}(0, 1)$ entries for the slow-decay case and i.i.d. $t(2)$ entries for the fast-decay case, and calculate $\boldsymbol{X} = \boldsymbol{\Sigma}^{1/2}\boldsymbol{Z}$. A noise vector $\boldsymbol{z}$ is similarly generated with i.i.d. $\mathcal{N}(0, 1)$ entries. We scale $\boldsymbol{\beta}$ and $\boldsymbol{\Sigma}$ to ensure that $\|\boldsymbol{\beta}\|_2 = c_1$ and $\|\boldsymbol{\Sigma}\|_F = c_2$ for $c_1 \in \{0, 1, 5\}$ and $c_2 \in \{100, 300\}$.

We present the Type I and Type II error rates in Table 2. We first note that all of the tests control the Type I error rate at $\alpha = 0.05$ reasonably well under $H_0$. In terms of the Type II error, the sketched $F$-test performs comparable to or better than CGZ and ZC tests in most of the considered scenarios. In particular, the sketched $F$-test has a clear advantage over the competitors for the fast-decay case, which coincides with our theory in Section 3.

**Asymptotic Behavior.** Our theoretical results regarding optimality of the proposed sketched $F$-test rely on the observation that $\Delta_k^2 / \boldsymbol{\beta}^\top \boldsymbol{\Sigma} \boldsymbol{\beta} \xrightarrow{\text{p}} 1$ within $\mathcal{D}(r)$ and $k = O(r)$. To illustrate the accuracy of this approximation, we consider the polynomial-decay condition of eigenvalues in Example 2 with the parameter $\alpha = 2$ and $\alpha = 4$. We then generate $\boldsymbol{\beta}$ and $\boldsymbol{U}$ in a similar manner to the previous simulation part, followed by a scaling step to ensure that $\boldsymbol{\beta}^\top \boldsymbol{\Sigma} \boldsymbol{\beta} = 1$; see more details in Figure 1.

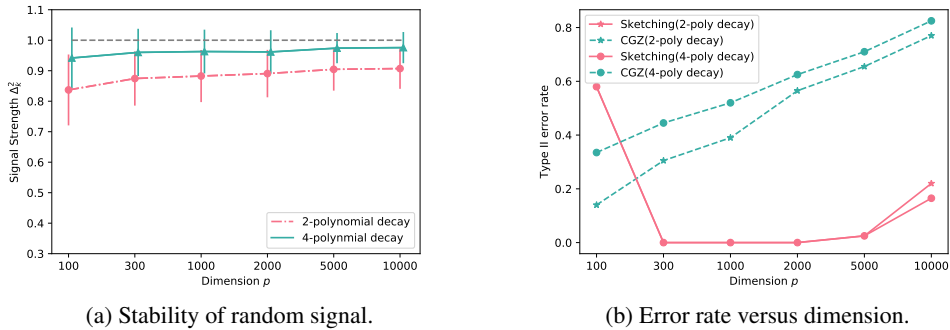

(a) Stability of random signal.  (b) Error rate versus dimension.

Figure 1: Simulation results with $p \in \{100, 300, 1000, 2000, 5000, 10000\}$ and $n = 10\log^2 p$. Results are based on 200 repeats. The first plot is generated with theoretical optimal $k = \lfloor \min\{3(\log p)^{\frac{1}{\alpha-1}}, n/2\} \rfloor$, while the second plot is based a practical choice of $k = \lfloor n/2 \rfloor$. The ZC test behaves similarly to CGZ but runs much slower, so we do not show detailed results here.

We evaluate the random signal strength in the noiseless setting before proceeding with synthetic data performance. With the optimal choice of $k$ suggested by Example 2, we plot the confidence intervals for $\Delta_k^2$ in Figure 1a. This empirical result confirms that the random-projection approach maintains robust signal strength, which is a building block of our theoretical analysis in Section 3. We then calculate the Type II error rates in a synthetic large $p$, small $n$ and non-sparse linear model with constant noise.

We make a note that in our simulations, we mainly consider cases where the eigenvalues of $\Sigma$ enjoy a decaying structure, in which settings, the designs are of intrinsically lower dimensions. For such decaying structures, the $U$-statistics type tests (e.g. CGZ, ZC) by design are not suitable, therefore are not as competitive. These simulation results support our theoretical findings, and show that, in this large $p$, small $n$ and non-sparse case where the global testing problem becomes extremely hard, our proposed random projection approach enjoys higher power.

## 5  Discussion

In this work, we consider the problem of testing the overall significance for the regression coefficients in the high-dimensional settings. Building upon the random projection techniques, we introduce a sketched $F$-test for arbitrary dimension and sample size pair and develop theoretical properties for the proposed test including the asymptotic power and minimax optimality. We also demonstrate the advantages of the proposed test over the existing competitors in terms of the asymptotic relative efficiency and computational complexity. To the best of our knowledge, the proposed procedure is the first attempt to analyze in details how sketching techniques work for testing regression coefficients.

Our findings and analysis suggest a few directions for further investigations. For example, our procedure, as a general methodology, can be substantially extended to other testing problems. For instance, built upon an improved argument of the high-dimensional $F$-test (see [35]), our framework can be *provably* adapted to testing whether $H_0 : G\boldsymbol{\beta} = r_0$ or $H_1 : G\boldsymbol{\beta} \neq r_0$ for matrix $G \in \mathbb{R}^{q \times p}$ and $r_0 \in \mathbb{R}^q$ with $q \leq p$. In the case where, the joint significance of a group of coefficients are tested, it is sensible to combine a sketching step (over the complement set of features) with the classical $F$-test. In addition, it would be interesting to see whether the sketching techniques can be applied to other types of tests, apart from the $F$-test, as an effective approach for dimension reduction and statistical inference.

**Broader Impact**   This work is a theoretical contribution to incorporate dimension reduction techniques (via random projections) to hypothesis testing in high dimensional regression. The insights from the proposed algorithm can potentially be leveraged in various hypothesis testing and machine learning tasks in the future.

## Footnotes

*Y. Wei is supported in part by the NSF grant CCF-2007911 and DMS-2015447.

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
