[Supplementary Material · supplementary.pdf]

# Supplementary file for "Randomized tests for high-dimensional regression: A more efficient and powerful solution"

## Abstract

This document provides the complete proofs and additional details for the main results stated in the NeurIPS submission titled "Randomized tests for high-dimensional regression: A more efficient and powerful solution".

## Contents

**Notation.** We use $\overset{d}{=}$ for two random variables that have the same distribution. Let $\Phi(\cdot)$ denote the CDF of $\mathcal{N}(0,1)$, and $z_\alpha$ denote the upper $\alpha$ quantile of $\mathcal{N}(0,1)$. The upper $\alpha$ quantile of $F$-distribution with degrees of freedom $(p, n-p)$ is denoted by $q_{\alpha,p,n-p}$. Moreover, the norm $\|\cdot\|_2$ stands for Euclidean norm for a vector, and spectral norm for a matrix. Matrix Frobenuis norm is denoted by $\|\cdot\|_F$. We call $a_n \asymp b_n$ if there is a universal constant $c_0$ such that $\frac{1}{c_0} \leq \frac{a_n}{b_n} \leq c_0$ for large enough $n$, and $a_n \lesssim b_n$ if $a_n \leq c_0 b_n$ for large enough $n$.

# A  Relaxation of Gaussian Assumptions

In this part, we show that the proposed sketching test is still valid under more general conditions for both data matrix and noise distribution. To do this, we invoke a new set of assumptions on $X_i$ and $z_i$ in model (2), which hold beyond the Gaussian setting.

**(B1)** The design vectors are generated as $x_i = \Gamma u_i$, where $\Gamma \in \mathbb{R}^{p \times m}$ satisfies $\Gamma \Gamma^\top = \Sigma$ and $u_1, \ldots, u_n$ are i.i.d. instances with $\mathbb{E}[u_i] = 0$ and $\mathrm{Var}[u_i] = I_m$ for some $m \leq k$. Additionally, we assume that $u_i$ satisfies

   **(a)** *(polynomial tail)* There exists constant $c, C > 0$ such that for any $n \in \mathbb{N}$, orthogonal projection $P$ in $\mathbb{R}^m$ and $t > C\mathrm{rank}(P)$, we have $\mathbb{P}(\|Pu_i\|^2 > t) \leq Ct^{-1-c}$;

   **(b)** *(bounded moment)* We have $\sup_{\|v\|=1}(\mathbb{E}|v'u_i|^8)^{1/8} = O(1)$ and for any symmetric matrix sequence $M \in \mathbb{R}^{m \times m}$,

$$\mathrm{Var}[u_i^\top M u_i] = O(\mathrm{tr}(M^2)) + o(\mathrm{tr}^2(M)).$$

**(B2)** The noise vector $z$ is independent of design matrix, with $\mathbb{E}[z_i^2] = 1$ and $\mathbb{E}[z_i^4] \leq c$ for $1 \leq i \leq n$ and some universal constant $c > 0$.

With this new set of assumptions, we are able to obtain similar results as in the Gaussian case. Theorem A.1 below, which builds on [7], includes Theorem 1 as a special case; we can also show Theorem 3 holds if we replace the Gaussian assumptions of $X$ and $z$ with **(B1)** and **(B2)**.

**Theorem A.1.** *Besides (B1) and (B2), assume* $\limsup k/n < 1$ *and* $\beta^\top \Sigma \beta = o(k/n)$. *Then, for almost all sequences of sketching matrix $S_k$, the power function $\Psi^S(S_k) = P\{F(S_k) > q_{\alpha,k,n-k}\}$ of test* (3) *satisfies*

$$\Psi_n^F - \Phi\left(-z_\alpha + \frac{\sqrt{n}\Delta_k^2}{\sigma^2}\sqrt{\frac{1-k/n}{2k/n}}\right) \to 0.$$

The proof of the result shares the same spirit as the proof of Theorem 1; one major difference is that, when the design matrix is not Gaussian, sketched noise $z_i^S$ is not independent of sketched data $S_k X_i$ anymore, requiring extra efforts to characterize the behavior of $F(S_k)$. We list some technical details in Section D.

**Remark:** We note that the assumptions **(B1)** and **(B2)** are mild. The moment and tail conditions hold for a wide range of random instances beyond Gaussian, including heavy-tailed ones such as log-normal. Also note that we do not require entries of $u_i$ to be independent to each other.

# B  Formal version of Definition 1

In the following, we present a full version of Definition 1, which accounts for more general scenarios.

**Definition B.1.** *We say model* (2) *has intrinsic dimension up to $r$, if we can find $\eta = o(1)$ and $r \leq p$, such that*

$$\left(\frac{1}{r}\sum_{i=1}^{r}\widetilde{\beta}_i^2\right) \cdot \left(\sum_{i=r+1}^{p}\lambda_i\right) + \sum_{i=r+1}^{p}\widetilde{\beta}_i^2\lambda_i \leq \eta\beta^\top\Sigma\beta;$$

$$\left(\frac{1}{r}\sum_{i=1}^{r}\widetilde{\beta}_i^2 + \frac{1}{p-r}\sum_{i=r+1}^{p}\widetilde{\beta}_i^2\right) \cdot r\lambda_{r+1} \leq \eta\beta^\top\Sigma\beta.$$

(1)

*Denote the collection of such $(\beta, \Sigma)$ as $\mathcal{D}(r)$.*

59 Here quantities $\eta$ and $r$ are both sequences of parameters indexed by $p$. When each $\widetilde{\beta}_i$ follows the
60 same distribution $\mathcal{P}$, the above conditions boil down to Definition 1.

## C  Proof of results in main text

### C.1  Proof of Theorem 1

63 In order to complete the proof, we need to check the conditions in Lemma 1 under the sketched
64 regression setting. For a fixed $S_k$, by the property of a conditional Gaussian distribution, we have

$$y_i|(\boldsymbol{X}_i'S_k) \sim N\left(\boldsymbol{\beta}^\top \boldsymbol{\Sigma} S_k (S_k^\top \boldsymbol{\Sigma} S_k)^{-1} S_k^\top \boldsymbol{X}_i, \nu^2\right),$$

65 with $\nu^2 := \sigma^2 + \boldsymbol{\beta}^\top \boldsymbol{\Sigma}\boldsymbol{\beta} - \Delta_k^2$. Additionally let us write $\boldsymbol{\beta}^S := (S_k^\top \boldsymbol{\Sigma} S_k)^{-1} S_k^\top \boldsymbol{\Sigma}\boldsymbol{\beta}$. Indeed
66 Algorithm 1 aims to test whether

$$H_0^S : \boldsymbol{\beta}^S = \boldsymbol{0} \quad \text{versus} \quad H_1^S : \boldsymbol{\beta}^S \neq \boldsymbol{0}, \tag{2}$$

67 for the new regression model

$$y_i = \boldsymbol{X}_i'S_k\boldsymbol{\beta}^S + z_i^S, \tag{3}$$

where $z_1^S, \ldots, z_n^S$ are independent random errors with $\text{Var}(z_i^S) = \nu^2$. Furthermore, when $S_k^\top \boldsymbol{\Sigma} S_k$ is
invertible, the problem stated in (2) becomes equivalent to testing whether

$$H_0^S : S_k^\top \boldsymbol{\Sigma}\boldsymbol{\beta} = 0 \quad \text{versus} \quad H_1^S : S_k^\top \boldsymbol{\Sigma}\boldsymbol{\beta} \neq 0.$$

68 It is shown in [6] that $\Delta_k^2 \leq \boldsymbol{\beta}^\top \boldsymbol{\Sigma}\boldsymbol{\beta}$. Then we can show $\Delta_k^2 = o(1)$ and $\nu^2 = \sigma^2 + o(1)$. Putting
69 pieces together with Lemma 1 completes the proof.

### C.2  Proof of Lemma 1

71 We present the full proof of Lemma 1 in this section. First, write the second term inside $\Phi(\cdot)$ as

$$\eta = \sqrt{\frac{(1-\delta)n}{2\delta}} \frac{\boldsymbol{\beta}^\top \boldsymbol{\Sigma}\boldsymbol{\beta}}{\sigma^2}. \tag{4}$$

72 We also define

$$\widehat{\sigma}^2 = \frac{\boldsymbol{y}^\top (\mathbf{I}_p - \boldsymbol{X}(\boldsymbol{X}^\top \boldsymbol{X})^{-1}\boldsymbol{X}^\top)\boldsymbol{y}}{n-p} \quad \text{and} \quad T = \frac{\widehat{\sigma}^2}{\sigma^2}\sqrt{\frac{n\delta(1-\delta)}{2}}(F-1).$$

73 The proof builds on the following two claims, which are proved at the end of this section.

$$\sqrt{n}\left(\frac{\widehat{\sigma}^2}{\sigma^2} - 1\right) = O_P(1) \quad \text{and} \tag{5}$$

$$T - \eta \xrightarrow{d} \mathcal{N}(0,1). \tag{6}$$

74 We now continue the main line of the proof assuming the claims in (5) and (6) hold. By the claim (5)
75 we know $\widehat{\sigma}^2/\sigma^2 \xrightarrow{P} 1$. Note that $\eta = o(\sqrt{n})$ under local alternative assumption. By Slutsky's
76 theorem,

$$G := \sqrt{\frac{n\delta(1-\delta)}{2}}(F-1) - \eta = \frac{\sigma^2}{\widehat{\sigma}^2}(T-\eta) + \left(\frac{\sigma^2}{\widehat{\sigma}^2} - 1\right)\eta \xrightarrow{d} \mathcal{N}(0,1). \tag{7}$$

77 We can use the convergence result (7) to show the claim in Lemma 1. Additionally write

$$s := \sqrt{\frac{n\delta(1-\delta)}{2}}(q_{\alpha,p,n-p} - 1). \tag{8}$$

78 Notice that $\Phi(\cdot)$ is Lipschitz-1 and thus we have

$$\left|\Psi_n^F - \Phi(-z_\alpha + \eta)\right| = \left|\mathbb{P}\left(G \geq s - \eta\right) - \Phi(-z_\alpha + \eta)\right|$$

$$\overset{(i)}{\leq} \left|\mathbb{P}\left(G \leq s - \eta\right) - \Phi\left(s - \eta\right)\right| + \left|\Phi\left(s - \eta\right) - \Phi(z_\alpha - \eta)\right|$$

$$\overset{(ii)}{\leq} \sup_{x\in\mathbb{R}}\left|\mathbb{P}\left(G \leq x\right) - \Phi\left(x\right)\right| + \left|s - z_\alpha\right|,$$

where step (i) uses the fact $\Phi(x) = 1 - \Phi(-x)$) and step (ii) uses Lipschitz property of $\Phi$. To analyze the second term, we need Lemma 2.1 of [1] which provides an approximation of $q_{\alpha,p,n-p}$ when $p = \delta n$ for $\delta \in (0,1)$.

**Lemma C.1** (Lemma 2.1 of [1]). *When $p = \delta n$ with $\delta \in (0,1)$, we have*

$$q_{\alpha,p,n-p} = 1 + \sqrt{\frac{2}{n\delta(1-\delta)}} z_\alpha + o(n^{-1/2}).$$

Rearranging the statement of Lemma C.1 yields $s = z_\alpha + o(1)$ where $s$ is defined in (8). We also know $\sup_{x\in\mathbb{R}} |\mathbb{P}(G \le x) - \Phi(x)| \to 0$ by the approximation (7). Combining these pieces yields $\left|\Psi_n^F - \Phi(-z_\alpha + \eta)\right| = o(1)$ and thus Lemma 1 follows.

**Proof of Claim** (5)

Write $\boldsymbol{H} = \boldsymbol{X}(\boldsymbol{X}^\top\boldsymbol{X})^{-1}\boldsymbol{X}^\top$. Notice that $\boldsymbol{H}\boldsymbol{X} = \boldsymbol{X}$ and then $(\mathbf{I}_p - \boldsymbol{H})\boldsymbol{X}\boldsymbol{\beta} = \mathbf{0}$. By the linearity assumption $\boldsymbol{y} = \boldsymbol{X}\boldsymbol{\beta} + \sigma\boldsymbol{z}$, we can write

$$\frac{\widehat{\sigma}^2}{\sigma^2} = \frac{(\boldsymbol{X}\boldsymbol{\beta} + \sigma\boldsymbol{z})^\top(\mathbf{I}_p - \boldsymbol{H})(\boldsymbol{X}\boldsymbol{\beta} + \sigma\boldsymbol{z})}{(n-p)\sigma^2} = \frac{1}{n-p}\boldsymbol{z}^\top(\mathbf{I}_p - \boldsymbol{H})\boldsymbol{z}. \tag{9}$$

Additionally, by our model assumption, the noise vector $\boldsymbol{z} \sim \mathcal{N}(\mathbf{0}, \mathbf{I}_p)$ is independent of $\boldsymbol{X}$. For any given $\boldsymbol{X}$ with rank $p$, $\mathbf{I}_p - \boldsymbol{H}$ is a projection matrix with rank $(n-p)$, and in this case $\boldsymbol{z}^\top(\mathbf{I}_p - \boldsymbol{H})\boldsymbol{z}|\boldsymbol{H} \sim \chi_{n-p}^2$. Under the Gaussian setting, we know $\text{rank}(\boldsymbol{X}) = p$ almost surely, so $\widehat{\sigma}^2/\sigma^2 \stackrel{d}{=} \chi_{n-p}^2/(n-p)$. Recall that $p = \delta n$, and thus $\sqrt{n}\left(\chi_{n-p}^2/(n-p) - 1\right) = O_P(1)$, which in turn leads to $\sqrt{n}\left(\widehat{\sigma}^2/\sigma^2 - 1\right) = O_P(1)$. This completes the proof of claim (5).

**Proof of Claim** (6)

We first rearrange the expression of $T$ in (6). By definition of $T$ in (6), we have

$$T = \frac{\widehat{\sigma}^2}{\sigma^2}\sqrt{\frac{n\delta(1-\delta)}{2}}(F-1) = \frac{\widehat{\sigma}^2}{\sigma^2}\sqrt{\frac{n\delta(1-\delta)}{2}}\left(\frac{\boldsymbol{y}^\top\boldsymbol{H}\boldsymbol{y}/p}{\widehat{\sigma}^2} - 1\right) = \sqrt{\frac{n\delta(1-\delta)}{2}}\left(\frac{\boldsymbol{y}^\top\boldsymbol{H}\boldsymbol{y}/p}{\sigma^2} - \frac{\widehat{\sigma}^2}{\sigma^2}\right).$$

Using the fact that $\boldsymbol{H}\boldsymbol{X} = \boldsymbol{X}$, we have

$$\boldsymbol{y}^\top\boldsymbol{H}\boldsymbol{y} = (\boldsymbol{X}\boldsymbol{\beta} + \sigma\boldsymbol{z})^\top\boldsymbol{H}(\boldsymbol{X}\boldsymbol{\beta} + \sigma\boldsymbol{z}) = \sigma^2\boldsymbol{z}^\top\boldsymbol{H}\boldsymbol{z} + 2\sigma\boldsymbol{\beta}^\top\boldsymbol{X}^\top\boldsymbol{z} + \boldsymbol{\beta}^\top\boldsymbol{X}^\top\boldsymbol{X}\boldsymbol{\beta}.$$

Combining the above with another expression of $\widehat{\sigma}^2/\sigma^2$ in (9), we can write $T$ as

$$T = \sqrt{\frac{n\delta(1-\delta)}{2}}\left(\frac{\boldsymbol{z}^\top\boldsymbol{H}\boldsymbol{z}}{p} - \frac{\boldsymbol{z}^\top(\mathbf{I}_p - \boldsymbol{H})\boldsymbol{z}}{n-p} + \frac{\boldsymbol{\beta}^\top\boldsymbol{X}^\top\boldsymbol{X}\boldsymbol{\beta}}{p\sigma^2} + \frac{2}{\sigma}\frac{\boldsymbol{\beta}^\top\boldsymbol{X}^\top\boldsymbol{z}}{p}\right).$$

By recalling $\eta$ defined in (4), we can decompose $T - \eta$ as $T - \eta = T_1 + (T_2 - \eta) + T_3$, where

$$T_1 = \sqrt{\frac{n\delta(1-\delta)}{2}}\left(\frac{\boldsymbol{z}^\top\boldsymbol{H}\boldsymbol{z}}{p} - \frac{\boldsymbol{z}^\top(\mathbf{I}_p - \boldsymbol{H})\boldsymbol{z}}{n-p}\right),$$

$$T_2 - \eta = \eta\left(\frac{\boldsymbol{\beta}^\top\boldsymbol{X}^\top\boldsymbol{X}\boldsymbol{\beta}}{n\boldsymbol{\beta}^\top\boldsymbol{\Sigma}\boldsymbol{\beta}} - 1\right) \quad \text{and}$$

$$T_3 = \frac{1}{\sigma}\sqrt{\frac{2(1-\delta)}{n\delta}}\boldsymbol{\beta}^\top\boldsymbol{X}^\top\boldsymbol{z}.$$

In what follows, we prove $T_1 \stackrel{d}{\to} \mathcal{N}(0,1)$, $T_2 - \eta \stackrel{d}{\to} 0$ and $T_3 \stackrel{d}{\to} 0$ and thus $T - \eta \stackrel{d}{\to} \mathcal{N}(0,1)$ as desired.

**Analyzing $T_1$:** Note that $\boldsymbol{H} = \boldsymbol{X}(\boldsymbol{X}^\top\boldsymbol{X})^{-1}\boldsymbol{X}^\top$ is a projection matrix with rank $p$ almost surely. Therefore, conditional on $\boldsymbol{H}$, we have $\boldsymbol{z}^\top\boldsymbol{H}\boldsymbol{z}|\boldsymbol{H} \sim \chi_p^2$ and $\boldsymbol{z}^\top(\boldsymbol{I} - \boldsymbol{H})\boldsymbol{z}|\boldsymbol{H} \sim \chi_{n-p}^2$ and these are

independent to each other. By letting $\omega_1, \omega_2 \overset{iid}{\sim} \mathcal{N}(0,1)$, we may apply the central limit theorem and see that

$$\boldsymbol{z}^\top \boldsymbol{H} \boldsymbol{z}/p | \boldsymbol{H} = 1 + \omega_1/\sqrt{p} + o_P(n^{-1/2}),$$

$$\boldsymbol{z}^\top (\boldsymbol{I} - \boldsymbol{H})\boldsymbol{z}/(n-p)|\boldsymbol{H} = 1 + \omega_2/\sqrt{n-p} + o_P(n^{-1/2}).$$

Then we conclude that $T_1 | \boldsymbol{H} \overset{d}{\to} \mathcal{N}(0,1)$ and thus $T_1 \overset{d}{\to} \mathcal{N}(0,1)$ as well by dominated convergence theorem.

**Analyzing $T_2$:** Since $\boldsymbol{X}\boldsymbol{\beta} \sim \mathcal{N}(\boldsymbol{0}, (\boldsymbol{\beta}^\top \boldsymbol{\Sigma} \boldsymbol{\beta}) \boldsymbol{I}_p)$ under the Gaussian setting, it follows that

$$\eta \left( \frac{\boldsymbol{\beta}^\top \boldsymbol{X}^\top \boldsymbol{X} \boldsymbol{\beta}}{n \boldsymbol{\beta}^\top \boldsymbol{\Sigma} \boldsymbol{\beta}} - 1 \right) \overset{d}{=} \eta \left( \frac{\chi_n^2}{n} - 1 \right).$$

Together with observations (i) $\eta = o(\sqrt{n})$ and (ii) $\sqrt{n}(\chi_n^2/n - 1) = O_P(1)$, we conclude $T_2 - \eta \overset{d}{\to} 0$.

**Analyzing $T_3$:** To show $T_3 \overset{d}{\to} 0$, it suffices to prove $\boldsymbol{\beta}^\top \boldsymbol{X}^\top \boldsymbol{z} = o_P(\sqrt{n})$. By the independence between $\boldsymbol{X}$ and $\boldsymbol{z}$, we have $\mathbb{E}\left[\boldsymbol{\beta}^\top \boldsymbol{X}^\top \boldsymbol{z}\right] = 0$ and $\mathrm{Var}(\boldsymbol{\beta}^\top \boldsymbol{X}^\top \boldsymbol{z}) = \mathbb{E}\left[\boldsymbol{\beta}^\top \boldsymbol{X}^\top \boldsymbol{z} \boldsymbol{z}^\top \boldsymbol{X} \boldsymbol{\beta}\right] = \mathbb{E}\left[\boldsymbol{\beta}^\top \boldsymbol{X}^\top \boldsymbol{X} \boldsymbol{\beta}\right] = n \boldsymbol{\beta}^\top \boldsymbol{\Sigma} \boldsymbol{\beta} = o(n)$. Therefore $\boldsymbol{\beta}^\top \boldsymbol{X}^\top \boldsymbol{z} = o_P(\sqrt{n})$ holds.

Combining the results, we complete the proof of claim (6).

## C.3 Well-definedness of Algorithm 1

In this part, we show that $S_k^\top \boldsymbol{X}^\top \boldsymbol{X} S_k$ is invertible almost surely.

Since $\mathrm{rank}(\boldsymbol{A}^\top \boldsymbol{A}) = \mathrm{rank}(\boldsymbol{A})$ for any matrix $\boldsymbol{A}$, we observe $\mathrm{rank}(S_k^\top \boldsymbol{X}^\top \boldsymbol{X} S_k) = \mathrm{rank}(\boldsymbol{X} S_k)$. For any realization of $\boldsymbol{X}$ with no all-zero rows, the entries of $\boldsymbol{X} S_k$ are independent Gaussian random variables and thus $\boldsymbol{X} S_k$ has full-rank $k$. By construction, $\boldsymbol{X}$ does not have all-zero rows almost surely, and thus $\mathrm{rank}(S_k^\top \boldsymbol{X}^\top \boldsymbol{X} S_k) = k$ almost surely.

## C.4 Proof of Proposition 1

Rearranging expression (8) in the main text, we have

$$\mathrm{ARE}_n(\Psi_n^{ZC}; \Psi_n^S) = \left( \frac{4}{\sqrt{\rho(1-\rho)}} \frac{\mathrm{tr}(\boldsymbol{\Sigma})}{\sqrt{\mathrm{tr}(\boldsymbol{\Sigma^2})}} \frac{1}{\sqrt{n}} \right) \cdot \left( \frac{\boldsymbol{\beta}^\top \boldsymbol{\Sigma} \boldsymbol{\beta}}{\Delta_k^2} \frac{k}{2p} \right) \cdot \left( \frac{\|\boldsymbol{\Sigma}\boldsymbol{\beta}\|^2}{\boldsymbol{\beta}^\top \boldsymbol{\Sigma} \boldsymbol{\beta}} \frac{p}{2\mathrm{tr}(\boldsymbol{\Sigma})} \right),$$

where we recall that

$$\Delta_k^2 := \boldsymbol{\beta}^\top \boldsymbol{\Sigma} S_k (S_k^\top \boldsymbol{\Sigma} S_k)^{-1} S_k^\top \boldsymbol{\Sigma} \boldsymbol{\beta}.$$

The first term is exactly what we want; it remains to derive high-probability bounds for the second and third terms. Define

$$\mathcal{E}_1 = \left\{ \frac{\Delta_k^2}{\boldsymbol{\beta}^\top \boldsymbol{\Sigma} \boldsymbol{\beta}} \geq \frac{k}{2p} \right\} \quad \text{and} \quad \mathcal{E}_2 = \left\{ \frac{\|\boldsymbol{\Sigma}\boldsymbol{\beta}\|^2}{\boldsymbol{\beta}^\top \boldsymbol{\Sigma} \boldsymbol{\beta}} \leq \frac{2\mathrm{tr}(\boldsymbol{\Sigma})}{p} \right\}.$$

If we can show $\mathbb{P}(\mathcal{E}_1) \to 1$ and $\mathbb{P}(\mathcal{E}_2) \to 1$ as $n \to \infty$, the claim of Proposition 1 follows.

The remaining parts of the proof rely on concentration bounds of Gaussian quadratic forms. See Lemma 0.2. in [2] for the proof of the following lemma:

**Lemma C.2** ([2]). *For any symmetric matrix $\boldsymbol{A} \in \mathbb{R}^{p \times p}$ with $\boldsymbol{A} \succeq 0$, $\boldsymbol{Z} \sim \mathcal{N}(0, I_{p \times p})$ and any $t > 0$, we have*

$$\mathbb{P}\left( \boldsymbol{Z}^\top \boldsymbol{A} \boldsymbol{Z} \geq \mathrm{tr}(\boldsymbol{A}) + 2\|\boldsymbol{A}\|_F \sqrt{t} + 2\|\boldsymbol{A}\|t \right) \leq \exp(-t) \quad and$$

$$\mathbb{P}\left( \boldsymbol{Z}^\top \boldsymbol{A} \boldsymbol{Z} \leq \mathrm{tr}(\boldsymbol{A}) - 2\|\boldsymbol{A}\|_F \sqrt{t} \right) \leq \exp(-t).$$

We also state the useful matrix inequality used in the proof:

**Lemma C.3.** *For a symmetric matrix $\boldsymbol{\Sigma} \in \mathbb{R}^{p \times p}$ and $\boldsymbol{\Sigma} \neq \mathbf{0}$, we have*

$$\frac{\mathrm{tr}(\boldsymbol{\Sigma})}{\|\boldsymbol{\Sigma}\|_F} \geq \left( \frac{\mathrm{tr}^2(\boldsymbol{\Sigma}^2)}{\mathrm{tr}(\boldsymbol{\Sigma}^4)} \right)^{1/8} .$$

129   The proof of Lemma C.3 can be found in Section D.1. Using Lemma C.2, we first show $\mathbb{P}(\mathcal{E}_1) \to 1$.

130   By assumption **(A)**, we can write $\boldsymbol{\Sigma}^{1/2}\boldsymbol{\beta}/\|\boldsymbol{\Sigma}^{1/2}\boldsymbol{\beta}\|_2$ as $\boldsymbol{Z}/\|\boldsymbol{Z}\|_2$, where $\boldsymbol{Z} \sim \mathcal{N}(\mathbf{0}, \boldsymbol{I}_p)$. Then

$$\frac{\Delta_k^2}{\boldsymbol{\beta}^\top \boldsymbol{\Sigma} \boldsymbol{\beta}} = \frac{1}{\|\boldsymbol{Z}\|_2^2} \boldsymbol{Z}^\top \boldsymbol{\Sigma}^{1/2} S_k (S_k^\top \boldsymbol{\Sigma} S_k)^{-1} S_k^\top \boldsymbol{\Sigma}^{1/2} \boldsymbol{Z} := \frac{1}{\|\boldsymbol{Z}\|_2^2} \boldsymbol{Z}^\top \boldsymbol{P} \boldsymbol{Z},$$

131   where we denote $\boldsymbol{P} := \boldsymbol{\Sigma}^{1/2} S_k (S_k^\top \boldsymbol{\Sigma} S_k)^{-1} S_k^\top \boldsymbol{\Sigma}^{1/2}$. To apply the second statement of
132   Lemma C.2, we first calculate $\mathrm{tr}(\boldsymbol{P})$ and $\|\boldsymbol{P}\|_F$. By $\mathrm{tr}(\boldsymbol{AB}) = \mathrm{tr}(\boldsymbol{BA})$, it follows that $\mathrm{tr}(\boldsymbol{P}) = $
133   $\mathrm{tr}((S_k^\top \boldsymbol{\Sigma} S_k)^{-1}(S_k^\top \boldsymbol{\Sigma} S_k)) = \mathrm{tr}(\boldsymbol{I}_k) = k$. Also notice that $\boldsymbol{P}$ is a projection matrix with rank $k$,
134   and then $\|\boldsymbol{P}\|_F = \sqrt{\mathrm{tr}(\boldsymbol{P}^\top \boldsymbol{P})} = \sqrt{\mathrm{tr}(\boldsymbol{P})} = \sqrt{k}$. By choosing $t = \frac{3-2\sqrt{2}}{8}k$, we have, for some
135   universal constant $C > 0$,

$$\mathbb{P}\left( \boldsymbol{Z}^\top \boldsymbol{P} \boldsymbol{Z} \leq \frac{k}{\sqrt{2}} \right) \leq \exp(-Ck).$$

136   By the law of large numbers, $\|\boldsymbol{Z}\|_2^2/p \to 1$ almost surely as $p \to \infty$. Thus $\mathbb{P}(\|\boldsymbol{Z}\|_2^2 \geq \sqrt{2}p) \to 0$.
137   By the above reasoning and the following lower bound

$$\mathbb{P}(\mathcal{E}_1) \geq 1 - \mathbb{P}\left( \|\boldsymbol{Z}\|_2^2 \geq \sqrt{2}p \right) + \mathbb{P}\left( \boldsymbol{Z}^\top \boldsymbol{P} \boldsymbol{Z} \leq \frac{k}{\sqrt{2}} \right),$$

138   we know $\mathbb{P}(\mathcal{E}_1) \to 1$ as $k \to \infty$ (recall that we assume $p \geq n/2$ and $k \to \infty$ as $n \to \infty$).

139   We complete the proof by showing $\mathbb{P}(\mathcal{E}_2) \to 1$. Similar to the proof in the first part, we may write

$$\frac{\|\boldsymbol{\Sigma}\boldsymbol{\beta}\|^2}{\boldsymbol{\beta}^\top \boldsymbol{\Sigma} \boldsymbol{\beta}} = \frac{1}{\|\boldsymbol{Z}\|^2} \boldsymbol{Z}^\top \boldsymbol{\Sigma} \boldsymbol{Z}.$$

140   Slightly modifying the first statement of Lemma C.2 yields

$$\mathbb{P}\left( \boldsymbol{Z}^\top \boldsymbol{\Sigma} \boldsymbol{Z} \geq \mathrm{tr}(\boldsymbol{\Sigma}) + 2\|\boldsymbol{\Sigma}\|_F \sqrt{t_1} + 2\|\boldsymbol{\Sigma}\|_2 t_2 \right) \leq \mathbb{P}\left( \boldsymbol{Z}^\top \boldsymbol{\Sigma} \boldsymbol{Z} \geq \mathrm{tr}(\boldsymbol{\Sigma}) + 2\|\boldsymbol{\Sigma}\|_F \sqrt{\min(t_1, t_2)} + 2\|\boldsymbol{\Sigma}\|_2 \min(t_1, t_2) \right)$$

$$\leq \exp(-\min(t_1, t_2)).$$

141   Choose $\sqrt{t_1} = \frac{\mathrm{tr}(\boldsymbol{\Sigma})}{24\|\boldsymbol{\Sigma}\|_F}$ and $t_2 = \frac{\mathrm{tr}(\boldsymbol{\Sigma})}{24\|\boldsymbol{\Sigma}\|_2}$. By $\|\boldsymbol{\Sigma}\|_F \geq \|\boldsymbol{\Sigma}\|_2$, we know $\sqrt{t_1} \leq t_2$. By Lemma C.3
142   and Condition (6), we observe $\sqrt{t_1} \to \infty$ as $p \to \infty$. Then

$$\mathbb{P}\left( \boldsymbol{Z}^\top \boldsymbol{\Sigma} \boldsymbol{Z} \geq \sqrt{2}\mathrm{tr}(\boldsymbol{\Sigma}) \right) \to 0, \quad p \to \infty.$$

143   Similar to the first part, we have

$$\mathbb{P}(\mathcal{E}_2) \geq 1 - \mathbb{P}(\|\boldsymbol{Z}\|^2 \geq \sqrt{2}p) - \mathbb{P}\left( \boldsymbol{Z}^\top \boldsymbol{\Sigma} \boldsymbol{Z} \geq \sqrt{2}\mathrm{tr}(\boldsymbol{\Sigma}) \right).$$

144   Recall that we have shown $\mathbb{P}(\|\boldsymbol{Z}\|^2 \geq \sqrt{2}p) \to 0$, and thus it follows that $\mathbb{P}(\mathcal{E}_2) \to 1$.

145   ## C.5  Details of Example 1

146   With the recommended choice $k = \lfloor n/2 \rfloor$, expression (9) in the main text becomes

$$\mathrm{ARE}_n(\Psi_n^{ZC}; \Psi_n^S) \leq 8 \frac{\mathrm{tr}(\boldsymbol{\Sigma})}{\sqrt{\mathrm{tr}(\boldsymbol{\Sigma^2})}} \frac{1}{\sqrt{n}}.$$

147   For Example 1, we have

$$\mathrm{tr}(\boldsymbol{\Sigma^2}) \geq \lambda_1^2 + \cdots + \lambda_s^2 \overset{(i)}{\geq} (\lambda_1 + \cdots + \lambda_s)^2/s \overset{(ii)}{\geq} (1-\epsilon)^2 \mathrm{tr}^2(\boldsymbol{\Sigma})/s,$$

148   where step (i) follows by Cauchy-Schwarz inequality and step (ii) uses the condition $\lambda_1 + \cdots + \lambda_s \geq$
149   $(1-\epsilon) \cdot \mathrm{tr}(\boldsymbol{\Sigma})$. This inequality further implies that

$$\mathrm{ARE}_n(\Psi_n^{ZC}; \Psi_n^S) \leq 8 \frac{\mathrm{tr}(\boldsymbol{\Sigma})}{\sqrt{\mathrm{tr}(\boldsymbol{\Sigma^2})}} \frac{1}{\sqrt{n}} \leq \frac{8\sqrt{s}}{(1-\epsilon)\sqrt{n}}.$$

150   With $s = \sqrt{n}$, we have $\frac{8\sqrt{s}}{(1-\epsilon)\sqrt{n}} \asymp n^{-1/4}$ and then $\mathrm{ARE}_n(\Psi_n^{ZC}; \Psi_n^S) \lesssim n^{-1/4}$.

## C.6  Proof of Theorem 2

The key of showing the upper bound part in Theorem 2 and Theorem 3 is a high-probability lower bound of the signal $\Delta_k^2$. Recall $\Delta_k^2 \leq \boldsymbol{\beta}^\top \boldsymbol{\Sigma} \boldsymbol{\beta}$. The Lemma C.4 below shows that, when $(\boldsymbol{\beta}, \boldsymbol{\Sigma}) \in \mathcal{D}(r)$ and sketching dimension is $O(r)$, the sketched model can capture most of signals in the original model.

**Lemma C.4.** *When $(\boldsymbol{\beta}, \boldsymbol{\Sigma}) \in \mathcal{D}(r)$ and $r \leq k = O(r)$, we have $\Delta_k^2 / \boldsymbol{\beta}^\top \boldsymbol{\Sigma} \boldsymbol{\beta} \xrightarrow{p} 1$.*

We defer the proof of Lemma C.4 to Section C.9. With the conclusion of Lemma C.4, we then establish Theorem 2 by first proving an information theoretic lower bound and then proving that our test achieves this lower bound.

### C.6.1  Lower bound

We start with the lower bound that is based on standard Le Cam's framework. Our argument is particularly similar to that in [3]. Without loss of generality, we assume $\sigma^2 = 1$. First, we define a new parameter class $B_r(\tau)$ as

$$B_r(\tau) = \left\{ \boldsymbol{\beta} \in \mathbb{R}^p : \|\boldsymbol{\beta}\|_2 \geq \tau, \beta_i = 0 \text{ for } r + 1 \leq i \leq p \right\}.$$

By definition of $\Theta_r(\tau)$, we can easily see that for any $\boldsymbol{\beta} \in B_r(\tau)$ and $\boldsymbol{\Sigma}_0 = \mathrm{diag}\,(\mathbf{1}_r, \mathbf{0}_{p-r})$, it follows $(\boldsymbol{\beta}, \boldsymbol{\Sigma}_0) \in \Theta_r(\tau)$. Then the minimax Type II error can be bounded by

$$\inf_{\psi} \sup_{\boldsymbol{\beta} \in B_r(\tau)} \mathbb{P}_{\boldsymbol{\beta}, \boldsymbol{\Sigma}_0}(\psi = 0) \leq \mathcal{R}_r(\tau).$$

Let $\mu$ be a probability measure on $B_r(\tau)$. Consider any family of probability measures $P_{\boldsymbol{\beta}}$ indexed by $\boldsymbol{\beta} \in B_r(\tau)$. Denote by $\mathbb{P}_\mu$ the mixture probability measure

$$\mathbb{P}_\mu = \int_{B_r(\tau)} P_{\boldsymbol{\beta}}\, \mu(d\boldsymbol{\beta}).$$

Also let $\chi^2(P', P) = \int (dP'/dP)^2 dP - 1$ be the chi-square divergence between two probability measures $P' \ll P$. Then,

$$\alpha + \mathcal{R}_r(\tau) \geq \inf_{\psi} \sup_{\boldsymbol{\beta} \in B_r(\tau)} \left\{ \mathbb{P}_0(\psi = 1) + \mathbb{P}_{\boldsymbol{\beta}, \boldsymbol{\Sigma}_0}(\psi = 0) \right\}$$

$$\geq 1 - \sqrt{\chi^2(\mathbb{P}_\mu, P_0)},$$

in which the infimum is taken over all test functions based on $(\boldsymbol{X}, \boldsymbol{y})$. To show the lower bound, it suffices to show that, for $\tau = \tau(A, n) = \frac{A r^{1/4}}{\sqrt{n}}$, we can find $\mu_\tau$ such that

$$\chi^2(\mathbb{P}_{\mu_\tau}, P_0) \leq 1 + o_A(1), \tag{10}$$

where $o_A(1)$ tends to 0 as $A \to 0$.

Note that when $\boldsymbol{\Sigma} = \boldsymbol{\Sigma}_0$, data matrix $\boldsymbol{X}$ under the null and alternative model only differs in the first $r$ features. Thus the chi-square divergence is essentially the divergence between two $r$-dimensional distributions, which allows us to borrow techniques for linear regression with $\boldsymbol{\Sigma} = \boldsymbol{I}_r$. More specifically, we may apply the results in Section 7.1 of [4] and observe that

$$\chi^2(\mathbb{P}_{\mu_\tau}, P_0) \leq \exp(A^2) \tag{11}$$

for some properly chosen $\mu_\tau$. See Section 7.1 of [4] or Section 4.4 of [3] for more details.

### C.6.2  Upper bound

We now turn to the upper bound. Recall that we always assume $\boldsymbol{\beta}^\top \boldsymbol{\Sigma} \boldsymbol{\beta} = O(1)$, since the problem is trivial otherwise. In order to show the upper bound, following the definition (ii), it suffices to show, if we choose $\psi^S$ to be the sketched $F$-test in Algorithm 1 associated with any fixed sequence of sketching matrix $\{S_k\} \in \mathcal{A}$, it holds that

$$r(\psi^S, \boldsymbol{\beta}_n, \boldsymbol{\Sigma}_n) = o_P(1), \quad \text{when} \quad \tau_n/\epsilon_n \to \infty \text{ and } (\boldsymbol{\beta}, \boldsymbol{\Sigma}) \in \Theta_r(\tau).$$

For $(\boldsymbol{\beta}, \boldsymbol{\Sigma}) \in \Theta_r(\tau)$, by Chebyshev inequality, we have

$$\mathbb{E}_{\boldsymbol{\beta}, \boldsymbol{\Sigma}} \left[ 1 - \psi^S \right] = \mathbb{P}(F^S > q_{\alpha, k, n-k}) \leq \frac{\mathrm{Var}_{\boldsymbol{\beta}, \boldsymbol{\Sigma}}(F^S)}{(q_{\alpha, k, n-k} - \mathbb{E}_{\boldsymbol{\beta}, \boldsymbol{\Sigma}} \left[ F^S \right])^2}. \tag{12}$$

We claim that the following inequalities hold, and leave their proofs to the end of this section:

$$\mathrm{Var}_{\boldsymbol{\beta}, \boldsymbol{\Sigma}}(F^S) \leq \frac{C}{r^2} \left[ \frac{r^2 + \lambda^2}{n} + (r + \lambda) \right] \quad \text{and} \tag{13}$$

$$(q_{\alpha, k, n-k} - \mathbb{E}_{\boldsymbol{\beta}, \boldsymbol{\Sigma}} \left[ F^S \right])^2 \geq \frac{\lambda^2}{2r^2}, \tag{14}$$

for any fixed $S_k \in \mathcal{A}$. Here we define $\lambda := n\Delta_k^2/\nu^2$ which satisfies $\sqrt{r}/\lambda = o_P(1)$. As a consequence of expression (13), we have

$$\mathbb{E}_{\boldsymbol{\beta}, \boldsymbol{\Sigma}} \left[ 1 - \psi^S \right] \leq C \frac{(r^2 + \lambda^2)/n + (r + \lambda)}{\lambda^2} = o_P(1).$$

This completes the proof.

**Proof of inequalities** (13) **and** (14)

We omit the subscript $\boldsymbol{\beta}$ and $\boldsymbol{\Sigma}$ of Var and $\mathbb{E}$ for short. Recall that we define $\boldsymbol{\beta}^S = (S_k^\top \boldsymbol{\Sigma} S_k)^{-1} S_k^\top \boldsymbol{\Sigma} \boldsymbol{\beta}$ and $\nu^2 = \sigma^2 + \boldsymbol{\beta}^\top \boldsymbol{\Sigma} \boldsymbol{\beta} - \Delta_k^2$. Following the reasoning in the proof of Theorem 1, we have under $H_1$,

$$F^S | \boldsymbol{X} \sim F_{k, n-k}(\lambda(\boldsymbol{X})) \quad \text{where} \quad \lambda(\boldsymbol{X}) = \frac{(\boldsymbol{\beta}^S)^\top S_k^\top \boldsymbol{X}^\top \boldsymbol{X} S_k \boldsymbol{\beta}^S}{\nu^2}.$$

By the moment expressions of a non-central $F$-statistic, it can be easily seen that

$$\mathbb{E}[F^S | \boldsymbol{X}] = \frac{(n - k)(k + \lambda(\boldsymbol{X}))}{k(n - k - 2)},$$

$$\mathrm{Var}(F^S | \boldsymbol{X}) = 2 \frac{(k + \lambda(\boldsymbol{X}))^2 + (k + 2\lambda(\boldsymbol{X}))(n - k - 2)}{(n - k - 2)^2 (n - k - 4)} \left( \frac{n - k}{k} \right)^2.$$

Then we have, with $\lambda := \mathbb{E}[\lambda(\boldsymbol{X})] = n\Delta_k^2/\nu^2$ and $\mathrm{Var}(\lambda(\boldsymbol{X})) = 2\lambda^2/n$,

$$\mathrm{Var}(\mathbb{E}[F^S | \boldsymbol{X}]) \leq \frac{2}{k^2} \mathrm{Var}(\lambda(\boldsymbol{X})),$$

$$\mathbb{E}[\mathrm{Var}(F^S | \boldsymbol{X})] \leq \frac{C}{k^2} \left[ \frac{(k + \lambda)^2}{n} + (k + \lambda) + \frac{\mathrm{Var}(\lambda(\boldsymbol{X}))}{n} \right].$$

By the law of total variance,

$$\mathrm{Var}(F^S) = \mathrm{Var}(\mathbb{E}[F^S | \boldsymbol{X}]) + \mathbb{E}[\mathrm{Var}(F^S | \boldsymbol{X})] \leq \frac{C}{k^2} \left[ \frac{(k + \lambda)^2}{n} + (k + \lambda) \right],$$

which proves inequality (13) under the assumption $k \asymp r$.

To prove inequality (14), notice that

$$\mathbb{E}[F^S] = \frac{(n - k)(k + \lambda)}{k(n - k - 2)}.$$

In addition, Lemma C.8. of [7] yields

$$q_{\alpha, k, n-k} = 1 + \sqrt{\frac{2n}{k(n - k)}} z_\alpha + o(k^{-1/2}).$$

By the assumption $\tau_n/\epsilon_n \to \infty$, it follows $\lambda \gg \sqrt{r}$. After checking each term in $(q_{\alpha, k, n-k} - \mathbb{E}[F^S])^2$, we have

$$(q_{\alpha, k, n-k} - \mathbb{E}[F^S])^2 = \frac{\lambda^2}{k^2}(1 + o(1)),$$

which verifies inequality (14).

## C.7 Proof of Theorem 3

By Theorem A.1 and Lemma C.4, it suffices to show that when $\Delta_k^2/\boldsymbol{\beta}^\top \boldsymbol{\Sigma} \boldsymbol{\beta} \xrightarrow{p} 1$, we have

$$y_n := \Phi\left(-z_\alpha + \frac{\sqrt{n}\Delta_k^2}{\sigma^2}\sqrt{\frac{1-k/n}{2k/n}}\right) - \Phi\left(-z_\alpha + \frac{\sqrt{n}\boldsymbol{\beta}^\top \boldsymbol{\Sigma}\boldsymbol{\beta}}{\sigma^2}\sqrt{\frac{1-k/n}{2k/n}}\right) \xrightarrow{p} 0. \quad (15)$$

For ease of notation, let us write $a_n = \frac{\sqrt{n}\boldsymbol{\beta}^\top \boldsymbol{\Sigma}\boldsymbol{\beta}}{\sigma^2}\sqrt{\frac{1-k/n}{2k/n}}$ and $\eta_n = \frac{\boldsymbol{\beta}^\top \boldsymbol{\Sigma}\boldsymbol{\beta} - \Delta_k^2}{\boldsymbol{\beta}^\top \boldsymbol{\Sigma}\boldsymbol{\beta}}$. Then we have $\eta_n \xrightarrow{p} 0$ and $\eta_n \geq 0$, due to the fact that $\Delta_k^2 \leq \boldsymbol{\beta}^\top \boldsymbol{\Sigma}\boldsymbol{\beta}$. Assume $n$ is large enough, such that $\eta_n \leq 1/2$. By Lipschitz-1 property of $\Phi(\cdot)$, we have

$$|y_n| \leq \eta_n a_n. \quad (16)$$

On the other hand, we have

$$|y_n| \overset{\text{(i)}}{\leq} \Phi\left(z_\alpha - \frac{\sqrt{n}\Delta_k^2}{\sigma^2}\sqrt{\frac{1-k/n}{2k/n}}\right) + \Phi\left(z_\alpha - \frac{\sqrt{n}\boldsymbol{\beta}^\top \boldsymbol{\Sigma}\boldsymbol{\beta}}{\sigma^2}\sqrt{\frac{1-k/n}{2k/n}}\right)$$

$$\leq 2\Phi\left(z_\alpha - \frac{\sqrt{n}\Delta_k^2}{\sigma^2}\sqrt{\frac{1-k/n}{2k/n}}\right) \quad (17)$$

$$\overset{\text{(ii)}}{\leq} 2\Phi(z_\alpha - a_n/2)$$

$$\overset{\text{(iii)}}{\leq} 2\exp\left\{-\frac{\mathbf{1}_{\{z_\alpha - a_n/2 \leq 0\}}(z_\alpha - a_n/2)^2}{2}\right\},$$

where step (i) is due to $\Phi(x) - \Phi(y) \leq |\Phi(x) - \Phi(y)| = |\Phi(-x) - \Phi(-y)| \leq \Phi(-x) + \Phi(-y)$, step (ii) follows from $\eta_n \leq 1/2$ and step (iii) uses the Gaussian tail bound $\Phi(x) \leq 2\exp(-\mathbf{1}_{\{x \leq 0\}}x^2/2)$.

Combining inequalities (16) and (17), we have

$$|y_n| \leq \min\left\{\eta_n a_n, 2\exp\left\{-\frac{\mathbf{1}_{\{z_\alpha - a_n/2 \leq 0\}}(z_\alpha - a_n/2)^2}{2}\right\}\right\}.$$

Given $\eta_n > 0$, we know $\eta_n a_n$ and $2\exp\left\{-\mathbf{1}_{\{z_\alpha - a_n/2 \leq 0\}}(z_\alpha - a_n/2)^2/2\right\}$ are monotone increasing and decreasing respectively as functions of $a_n$. Then we have the upper bound

$$|y_n| \leq 2\exp\left\{-\frac{\mathbf{1}_{\{z_\alpha - f(\eta_n)/2 \leq 0\}}(z_\alpha - f(\eta_n)/2)^2}{2}\right\}, \quad (18)$$

where $f(\eta_n)$ is the unique $x_n$ that solves

$$\eta_n x_n = 2\exp\left\{-\frac{\mathbf{1}_{\{z_\alpha - x_n/2 \leq 0\}}(z_\alpha - x_n/2)^2}{2}\right\}.$$

We can directly check that $f(\eta_n)$ is a monotone decreasing function of $\eta_n$, and $\lim_{\eta_n \to 0^+} f(\eta_n) = +\infty$. Then

$$\lim_{\eta_n \to 0^+} 2\exp\left\{-\frac{\mathbf{1}_{\{z_\alpha - f(\eta_n)/2 \leq 0\}}(z_\alpha - f(\eta_n)/2)^2}{2}\right\} = 0.$$

By bound (18), it follows that $y_n \xrightarrow{p} 0$. This completes the proof of Theorem 3.

## C.8 Details of Examples 2

In this section, we provide details of Examples 2 with $\eta = 1/\log p$. Note that for the first two cases, the conditions in Definition B.1 essentially boil down to that in (10) in the main text.

For the $\alpha$-polynomial decay case, notice that

$$\sum_{i=r+1}^{p} \lambda_i \asymp r^{-\alpha+1}, \quad \sum_{i=1}^{p} \lambda_i \asymp 1.$$

Then the conditions translate to $r^{-\alpha+1} \leq 1/\log p$, or equivalently, there exists $r \lesssim (\log p)^{\frac{1}{\alpha-1}}$ such that $(\boldsymbol{\beta}, \boldsymbol{\Sigma}) \in \mathcal{D}(r)$.

For the $\gamma$-exponential decay example, first note that $\lambda_k/\lambda_{k+1} = \exp((k+1)^\gamma - k^\gamma)$ by definition. When $\gamma \geq 1$, we have $\lambda_k/\lambda_{k+1} \geq e \geq 1 + 1/k$; whereas $0 < \gamma < 1$, we have

$$(k+1)^\gamma - k^\gamma = \gamma \int_k^{k+1} x^{\gamma-1} dx \geq \gamma(k+1)^{\gamma-1} \geq \frac{\gamma}{k}.$$

Thus $\lambda_k/\lambda_{k+1} \geq 1 + \gamma/k$. In either case, we know $\{\lambda_k\}$ decays faster than $(\gamma \wedge 1)/k$. Then observe that

$$\sum_{i=r+1}^p \lambda_i \leq \lambda_r \sum_{i=r+1}^p \frac{\gamma \wedge 1}{i} \lesssim \exp(-r^\gamma)(\log p), \quad \sum_{i=1}^p \lambda_i \asymp 1.$$

Thus the conditions translate to $\exp(-r^\gamma)(\log p) \leq 1/\log p$ and $r \exp(-r^\gamma) \leq 1/\log p$, or there exists $r \lesssim (\log \log p)^{\frac{1}{\gamma}}$ such that $(\boldsymbol{\beta}, \boldsymbol{\Sigma}) \in \mathcal{D}(r)$ as stated in Table 1 in the main file.

For the structured coefficient example, we know that

$$\sum_{i=1}^r \widetilde{\beta}_i \asymp \log r, \quad \sum_{i=r+1}^p \lambda_i \asymp \log p - \log r, \quad \sum_{i=r+1}^p \widetilde{\beta}_i^2 \lambda_i \asymp 1/r, \quad \boldsymbol{\beta}^\top \boldsymbol{\Sigma} \boldsymbol{\beta} \asymp 1.$$

Then the first condition of Definition B.1 is now $\log r(\log p - \log r)/r + 1/r \leq 1/\log p$, or $r \geq \log^2 p \log r$. Thus we can see that there exists $r \lesssim (\log p)^3$ satisfying both conditions.

## C.9 Proof of Lemma C.4

First we introduce some additional notation. In the SVD decomposition $\boldsymbol{\Sigma} = \boldsymbol{U}\boldsymbol{\Lambda}\boldsymbol{U}^\top$, write $\boldsymbol{U} = [\boldsymbol{U}_r \ \boldsymbol{U}_{p-r}]$ and $\boldsymbol{\Lambda} = \begin{bmatrix} \boldsymbol{\Lambda}_r & \\ & \boldsymbol{\Lambda}_{p-r} \end{bmatrix}$, where $\boldsymbol{U}_r \in \mathbb{R}^{p \times r}$ and $\boldsymbol{\Lambda}_r \in \mathbb{R}^{r \times r}$. Then $\boldsymbol{\Sigma} = \boldsymbol{U}_r \boldsymbol{\Lambda}_r \boldsymbol{U}_r^\top + \boldsymbol{U}_{p-r} \boldsymbol{\Lambda}_{p-r} \boldsymbol{U}_{p-r}^\top := \boldsymbol{\Sigma}_r + \boldsymbol{\Sigma}_{p-r}$.

The intuition comes from low-rank case. If $\mathrm{rank}(\boldsymbol{\Sigma}) = r$, using sketching dimension $k = r$ is enough. To see this, notice that when $\mathrm{rank}(\boldsymbol{\Sigma}) = r$, we have $\mathrm{rank}(\boldsymbol{\Sigma} S_k) = r$ almost surely, i.e., $\boldsymbol{\Sigma} S_k$ is of full-rank almost surely. Then $\exists \boldsymbol{\xi} \in \mathbb{R}^p$, such that $\boldsymbol{\Sigma}\boldsymbol{\beta} = \boldsymbol{\Sigma} S_k \boldsymbol{\xi}$. It follows that $\Delta_k^2 = \boldsymbol{\beta}^\top \boldsymbol{\Sigma} S_k (S_k^\top \boldsymbol{\Sigma} S_k)^{-1} S_k^\top \boldsymbol{\Sigma} S_k \boldsymbol{\xi} = \boldsymbol{\beta}^\top \boldsymbol{\Sigma} \boldsymbol{\beta}$.

In general case, we may not be able to find $\boldsymbol{\xi}$ satisfying $\boldsymbol{\Sigma}\boldsymbol{\beta} = \boldsymbol{\Sigma} S_k \boldsymbol{\xi}$, and we seek for some $\boldsymbol{\xi}$ to make the difference between $\boldsymbol{\Sigma}\boldsymbol{\beta}$ and $\boldsymbol{\Sigma} S_k \boldsymbol{\xi}$ small. Formally, as long as sketching dimension $k \geq r$, for any $\boldsymbol{\xi}$ that satisfies

$$\boldsymbol{U}_r^\top \boldsymbol{\beta} = \boldsymbol{U}_r^\top S_k \boldsymbol{\xi}, \tag{19}$$

define $\boldsymbol{\nu} = \boldsymbol{\Sigma}_{p-r}(\boldsymbol{\beta} - S_k \boldsymbol{\xi})$. Then $\boldsymbol{\Sigma}\boldsymbol{\beta} = \boldsymbol{\Sigma} S_k \boldsymbol{\xi} + \boldsymbol{\nu}$, and

$$\begin{aligned} \Delta_k^2 &= (\boldsymbol{\xi}^\top S_k^\top \boldsymbol{\Sigma} + \boldsymbol{\nu}^\top) S_k (S_k^\top \boldsymbol{\Sigma} S_k)^{-1} S_k^\top (\boldsymbol{\Sigma} S_k \boldsymbol{\xi} + \boldsymbol{\nu}) \\ &\geq \boldsymbol{\beta}^\top \boldsymbol{\Sigma} \boldsymbol{\beta} - (\boldsymbol{\beta} - S_k \boldsymbol{\xi})^\top \boldsymbol{\Sigma}_{p-r}(\boldsymbol{\beta} - S_k \boldsymbol{\xi}) \\ &\geq \boldsymbol{\beta}^\top \boldsymbol{\Sigma} \boldsymbol{\beta} - (\boldsymbol{\beta} - S_k \boldsymbol{\xi})^\top \boldsymbol{\Sigma}_{p-r}^+ (\boldsymbol{\beta} - S_k \boldsymbol{\xi}) \end{aligned}$$

for any $\boldsymbol{\Sigma}_{p-r}^+ \succeq \boldsymbol{\Sigma}_{p-r}$, where the first inequality follows by positive semi-definite property of $S_k(S_k^\top \boldsymbol{\Sigma} S_k)^{-1} S_k^\top$. When $k \geq r$, we have $\mathrm{rank}(\boldsymbol{U}_r^\top S_k) = r$ almost surely, so such $\boldsymbol{\xi}$ exists. To optimize the results, we seek for a solution of the problem

$$\min_{\boldsymbol{\xi}} (\boldsymbol{\beta} - S_k \boldsymbol{\xi})^\top \boldsymbol{\Sigma}_{p-r}^+ (\boldsymbol{\beta} - S_k \boldsymbol{\xi}) \quad s.t. \quad \boldsymbol{U}_r^\top (\boldsymbol{\beta} - S_k \boldsymbol{\xi}) = 0.$$

The optimal $\boldsymbol{\xi}^*$ can be obtained by Lagrange multiplier. With Lagrange function

$$\mathcal{L}(\boldsymbol{\xi}, \boldsymbol{\lambda}) = \frac{1}{2}(\boldsymbol{\beta} - S_k \boldsymbol{\xi})^\top \boldsymbol{\Sigma}_{p-r}^+ (\boldsymbol{\beta} - S_k \boldsymbol{\xi}) - \boldsymbol{\lambda}^\top \boldsymbol{U}_r^\top (\boldsymbol{\beta} - S_k \boldsymbol{\xi}),$$

by solving the following two equations

$$\frac{\partial \mathcal{L}(\boldsymbol{\xi}, \boldsymbol{\lambda})}{\partial \boldsymbol{\lambda}} = 0 \quad \text{and} \quad \boldsymbol{U}_r^\top (\boldsymbol{\beta} - S_k \boldsymbol{\xi}) = 0, \tag{20}$$

we can solve for $\boldsymbol{\xi}^*$. The following lemma gives an upper bound on $(\boldsymbol{\beta} - S_k \boldsymbol{\xi}^*)^\top \boldsymbol{\Sigma}_{p-r}^+ (\boldsymbol{\beta} - S_k \boldsymbol{\xi}^*)$:

**Lemma C.5.** *Write* $\widetilde{S}_1 = \boldsymbol{U}_r^\top S_k$, $\widetilde{S}_2 = \boldsymbol{U}_{p-r}^\top S_k$, $\widetilde{\boldsymbol{\beta}}_1 = \boldsymbol{U}_r^\top \boldsymbol{\beta}$ *and* $\widetilde{\boldsymbol{\beta}}_2 = \boldsymbol{U}_{p-r}^\top \boldsymbol{\beta}$. *With* $\boldsymbol{\xi}^*$ *in* (20), *we have* $(\boldsymbol{\beta} - S_k\boldsymbol{\xi}^*)^\top \boldsymbol{\Sigma}_{p-r}^+ (\boldsymbol{\beta} - S_k\boldsymbol{\xi}^*) \leq L_1 + L_2$, *where*

$$
\begin{aligned}
L_1 &= \frac{\|\widetilde{S}_2^\top \boldsymbol{\Lambda}_{p-r}^+ \widetilde{S}_2\|_2}{\lambda_{\min}(\widetilde{S}_1 \widetilde{S}_1^\top)} \|\widetilde{\boldsymbol{\beta}}_1\|_2^2, \\
L_2 &= \left(1 + \kappa(\widetilde{S}_2^\top \boldsymbol{\Lambda}_{p-r}^+ \widetilde{S}_2)\kappa(\widetilde{S}_1 \widetilde{S}_1^\top)\right) \cdot \widetilde{\boldsymbol{\beta}}_2^\top \boldsymbol{\Lambda}_{p-r}^+ \widetilde{\boldsymbol{\beta}}_2.
\end{aligned}
\tag{21}
$$

*Here* $\kappa(\cdot)$ *represents the condition number of matrix, i.e.,* $\kappa(\boldsymbol{A}) = \lambda_{\max}(\boldsymbol{A})/\lambda_{\min}(\boldsymbol{A})$.

To analyze the terms on the right hand side of (21), we make use of the following lemma.

**Lemma C.6.** *For* $\forall a > 1$, *if we choose sketching dimension to be* $ar \leq k \leq C_1 \frac{\sum_{i=r+1}^p \lambda_i}{\lambda_{r+1}}$, *then with probability at least* $1 - \exp(-c_2 r) - \exp(-c_1 \frac{\sum_{i=r+1}^p \lambda_i}{\lambda_{r+1}})$, *we obtain*

       *1.* $\kappa(\widetilde{S}_2^\top \Lambda_{p-r} \widetilde{S}_2) \leq 4$;

       *2.* $\lambda_{\max}(\widetilde{S}_2^\top \Lambda_{p-r} \widetilde{S}_2) \leq 2 \sum_{i=r+1}^p \lambda_i$;

       *3.* $\kappa(\widetilde{S}_1 \widetilde{S}_1^\top) \leq C_2$;

       *4.* $\lambda_{\min}(\widetilde{S}_1 \widetilde{S}_1^\top) \geq C_2^{-1} k$,

*where* $c_1, c_2, C_1, C_2$ *are universal constants only depending on* $a$.

The proof of Lemma C.6 can be found at the end of this section. Now suppose that Lemma C.6 is given and also assume that $ar \leq k \leq br$ with $a > 1$.

By Lemma C.5, we have

$$
\boldsymbol{\beta}^\top \boldsymbol{\Sigma} \boldsymbol{\beta} - \Delta_k^2 \leq \frac{2\|\widetilde{S}_2^\top \boldsymbol{\Lambda}_{p-r}^+ \widetilde{S}_2\|_2}{\lambda_{\min}(\widetilde{S}_1 \widetilde{S}_1^\top)} \|\widetilde{\boldsymbol{\beta}}_1\|_2^2 + 2\left(1 + \kappa(\widetilde{S}_2^\top \boldsymbol{\Lambda}_{p-r}^+ \widetilde{S}_2)\kappa(\widetilde{S}_1 \widetilde{S}_1^\top)\right) \cdot \widetilde{\boldsymbol{\beta}}_2^\top \boldsymbol{\Lambda}_{p-r}^+ \widetilde{\boldsymbol{\beta}}_2.
$$

Here we write $\boldsymbol{\Sigma}_{p-r}^+ = \boldsymbol{U}_{p-r}^\top \boldsymbol{\Lambda}_{p-r}^+ \boldsymbol{U}_{p-r}$, with $\boldsymbol{\Lambda}_{p-r}^+ = \mathrm{diag}(\lambda_{r+1}^+, \dots, \lambda_p^+)$. Then by Lemma C.7, when sketching dimension $k$ satisfies $ar \leq k \leq C_1 \sum_{i=r+1}^p \lambda_i^+ / \lambda_{r+1}^+$, we have

$$
\begin{aligned}
\boldsymbol{\beta}^\top \boldsymbol{\Sigma} \boldsymbol{\beta} - \Delta_k^2 &\leq \frac{4C_2 \sum_{i=r+1}^p \lambda_i^+}{k} \|\widetilde{\boldsymbol{\beta}}_1\|_2^2 + 2(1 + 4C_2) \cdot \widetilde{\boldsymbol{\beta}}_2^\top \boldsymbol{\Lambda}_{p-r}^+ \widetilde{\boldsymbol{\beta}}_2 \\
&\leq C_3 \left(\left(\frac{1}{r}\sum_{i=1}^r \widetilde{\beta}_i^2\right) \cdot \left(\sum_{i=r+1}^p \lambda_i^+\right) + \sum_{i=r+1}^p \widetilde{\beta}_i^2 \lambda_i^+\right)
\end{aligned}
\tag{22}
$$

with probability at least $1 - \exp(-c_2 r) - \exp(-c_1 \sum_{i=r+1}^p \lambda_i^+ / \lambda_{r+1}^+)$. Note that the constant $C_3$ here only depends on $a$.

Up to now, the derivations do not depend on the form of matrix $\boldsymbol{\Sigma}_{p-r}^+$. Now we are ready to choose a particular form of $\boldsymbol{\Sigma}_{p-r}^+$, namely we can set

$$
\boldsymbol{\Sigma}_{p-r}^+ := \boldsymbol{U}_{p-r}^\top \boldsymbol{\Lambda}_{p-r}^+ \boldsymbol{U}_{p-r} \quad \text{and} \quad \lambda_i^+ = \lambda_i + \frac{b}{C_1} \frac{r\lambda_{r+1}}{p-r}, \quad \text{for } r+1 \leq i \leq p.
\tag{23}
$$

Then direct calculations give

$$
\frac{\sum_{i=r+1}^p \lambda_i^+}{\lambda_{r+1}^+} = \frac{\sum_{i=r+1}^p \lambda_i + \frac{b}{C_1} r\lambda_{r+1}}{\lambda_{r+1} + \frac{b}{C_1}\frac{r\lambda_{r+1}}{p-r}} \geq \frac{b}{C_1} r.
$$

Plugging the expression of $\lambda_i^+$ into (22) and then applying the conditions in Definition B.1 yield the following result:

When sketching dimension $k$ satisfies $ar \leq k \leq br$, we have with probability at least $1 - 2\exp(-c_3 r)$ that

$$1 - \frac{\Delta_k^2}{\boldsymbol{\beta}^\top \boldsymbol{\Sigma} \boldsymbol{\beta}} \leq C_3 \left(1 + \frac{b}{C_1}\right) \eta = o(1).$$

Thus we finish the proof with the stronger conclusion $\frac{\Delta_k^2}{\boldsymbol{\beta}^\top \boldsymbol{\Sigma} \boldsymbol{\beta}} \xrightarrow{P} 1$.

Now we are only left to prove Lemma C.6.

**Proof of Lemma C.6.** To establish Lemma C.6, we make use of the result below whose proof is provided in Section D.1.

**Lemma C.7.** *Suppose* $\boldsymbol{\Lambda} = \mathrm{diag}(\lambda_1, \ldots, \lambda_N)$ *with* $\lambda_i \geq 0$, $\|\lambda\|^2 > 0$ *and* $S \in \mathbb{R}^{N \times n}$ *is a standard Gaussian random matrix with* $n \leq N$. *Write* $\boldsymbol{\lambda} = (\lambda_1, \ldots, \lambda_N)$. *Then for* $t < 1$,

$$(1-t)\sqrt{\sum_{i=1}^N \lambda_i^2} \leq s_{\min}(\boldsymbol{\Lambda} S) \leq s_{\max}(\boldsymbol{\Lambda} S) \leq (1+t)\sqrt{\sum_{i=1}^N \lambda_i^2},$$

*with probability at least* $1 - 9^n \cdot 2\exp\left(-\min\left\{\frac{1}{16}\frac{\|\lambda\|_2^4}{\|\lambda\|_4^4} t^2, \frac{1}{4}\frac{\|\lambda\|_2^2}{\|\lambda\|_\infty^2} t\right\}\right)$.

Applying Lemma C.7 to $\boldsymbol{\Lambda}_{p-r}^{1/2}\widetilde{S}_2$ with sketching dimension $k$ and $t = 1/3$ yields

$$\kappa(\widetilde{S}_2^\top \boldsymbol{\Lambda}_{p-r}\widetilde{S}_2) \leq 4 \quad \text{and} \quad \lambda_{\max}(\widetilde{S}_2^\top \boldsymbol{\Lambda}_{p-r}\widetilde{S}_2) \leq 2\sum_{i=r+1}^p \lambda_i \tag{24}$$

with probability at least $1 - \exp\left(\ln 9 \cdot k - \min\left\{\frac{1}{144}\frac{(\sum_{i=r+1}^p \lambda_i)^2}{\sum_{i=r+1}^p \lambda_i^2}, \frac{1}{12}\frac{\sum_{i=r+1}^p \lambda_i}{\lambda_{r+1}}\right\}\right)$. Since $\frac{(\sum_{i=r+1}^p \lambda_i)^2}{\sum_{i=r+1}^p \lambda_i^2} \geq \frac{\sum_{i=r+1}^p \lambda_i}{\lambda_{r+1}}$, (24) holds with probability at least $1 - \exp\left(-c_1 \frac{\sum_{i=r+1}^p \lambda_i}{\lambda_{r+1}}\right)$ as long as $k \leq C_1 \frac{\sum_{i=r+1}^p \lambda_i}{\lambda_{r+1}}$.

Lemma C.7 can be used to bound all the four quantities in Lemma C.6. To obtain a better control for $\kappa(\widetilde{S}_1\widetilde{S}_1^\top)$ and $\lambda_{\min}(\widetilde{S}_1\widetilde{S}_1^\top)$ in terms of constants, we invoke the following lemma from [6]:

**Lemma C.8** (Lemma 4 of [6]). *For* $k \leq p$, *let* $P_k \in \mathbb{R}^{k \times p}$ *be a random matrix with i.i.d.* $\mathcal{N}(0,1)$ *entries. Then*

$$\mathbb{P}\left(\lambda_{\max}(\frac{1}{p}P_k^\top P_k) \geq (1 + \sqrt{k/p} + t)^2\right) \leq \exp(-pt^2/2);$$

$$\mathbb{P}\left(\lambda_{\min}(\frac{1}{p}P_k^\top P_k) \leq (1 - \sqrt{k/p} - t)^2\right) \leq \exp(-pt^2/2).$$

With constant $a > 1$ and $k \geq ar$, we now apply Lemma C.8 to $\widetilde{S}_1$ and obtain that with probability at least $1 - \exp(-c_2 r)$,

$$\kappa(\widetilde{S}_1\widetilde{S}_1^\top) \leq C_2 \quad \text{and} \quad \lambda_{\min}(\widetilde{S}_1\widetilde{S}_1^\top) \geq C_2^{-1}k \tag{25}$$

where $c_2, C_2$ are universal constants only depending on $a$. This completes the proof of Lemma C.6.

## C.10 Proof of Lemma C.5

**Structure of the proof:** We prove Lemma C.5 following the Lagrange multiplier procedure discussed in the main text. We first derive the expression of $\boldsymbol{\xi}^*$ using the Lagrange multiplier; the explicit form of $\boldsymbol{\xi}^*$ is summarized in (26) and (27). Then we plug $\boldsymbol{\xi}^*$ into $(\boldsymbol{\beta} - S_k\boldsymbol{\xi}^*)^\top \boldsymbol{\Sigma}_{p-r}(\boldsymbol{\beta} - S_k\boldsymbol{\xi}^*)$, and get its upper bound; see (28). The remaining part of the proof proceeds by bounding the terms in (28) based on properties of the spectral norm.

**Step 1: Finding minimal value of** $(\boldsymbol{\beta} - S_k\boldsymbol{\xi}^*)^\top \boldsymbol{\Sigma}_{p-r}(\boldsymbol{\beta} - S_k\boldsymbol{\xi}^*)$**.**

Recall that we define the Lagrange form

$$\mathcal{L}(\boldsymbol{\xi}, \boldsymbol{\lambda}) = \frac{1}{2}(\boldsymbol{\beta} - S_k\boldsymbol{\xi})^\top \boldsymbol{\Sigma}_{p-r}(\boldsymbol{\beta} - S_k\boldsymbol{\xi}) - \boldsymbol{\lambda}^\top \boldsymbol{U}_r^\top(\boldsymbol{\beta} - S_k\boldsymbol{\xi}).$$

By solving the following two equations

$$\frac{\partial \mathcal{L}(\boldsymbol{\xi}, \boldsymbol{\lambda})}{\partial \boldsymbol{\xi}} = 0 \quad \text{and} \quad \boldsymbol{U}_r^\top (\boldsymbol{\beta} - S_k \boldsymbol{\xi}) = 0,$$

we can obtain the optimal solution $\boldsymbol{\xi}^*$.

First, let us consider the first equation

$$\frac{\partial \mathcal{L}(\boldsymbol{\xi}, \boldsymbol{\lambda})}{\partial \boldsymbol{\xi}} = 0.$$

A direct calculation yields

$$S_k^\top \boldsymbol{\Sigma}_{p-r} S_k \boldsymbol{\xi} - S_k^\top \boldsymbol{\Sigma}_{p-r} \boldsymbol{\beta} + S_k^\top \boldsymbol{U}_r \boldsymbol{\lambda} = 0.$$

Similar to proof in Section C.3 and by noting that $\text{rank}(\boldsymbol{\Sigma}_{p-r}) = p - r \geq k$, we can show the matrix $S_k^\top \boldsymbol{\Sigma}_{p-r} S_k$ is invertible almost surely. Then the solution can be written explicitly as

$$\boldsymbol{\xi}^* = (S_k^\top \boldsymbol{\Sigma}_{p-r} S_k)^{-1} S_k^\top (\boldsymbol{\Sigma}_{p-r} \boldsymbol{\beta} - \boldsymbol{U}_r \boldsymbol{\lambda}^*). \tag{26}$$

By writing $\boldsymbol{H} = S_k(S_k^\top \boldsymbol{\Sigma}_{p-r} S_k)^{-1} S_k^\top$ and plugging the above expression to the constraint condition $\boldsymbol{U}_r^\top (\boldsymbol{\beta} - S_k \boldsymbol{\xi}) = 0$, we obtain the following equality:

$$\boldsymbol{U}_r^\top \boldsymbol{\beta} - \boldsymbol{U}_r^\top \boldsymbol{H} (\boldsymbol{\Sigma}_{p-r} \boldsymbol{\beta} - \boldsymbol{U}_r \boldsymbol{\lambda}^*) = 0.$$

Before preceding, we first justify that $\boldsymbol{U}_r^\top \boldsymbol{H} \boldsymbol{U}_r$ is invertible almost surely. Note that when $S_k^\top \boldsymbol{\Sigma}_{p-r} S_k$ is invertible, we have $\boldsymbol{x}^\top \boldsymbol{U}_r^\top \boldsymbol{H} \boldsymbol{U}_r \boldsymbol{x} = 0$ iff $S_k^\top \boldsymbol{U}_r \boldsymbol{x} = 0$ iff $\boldsymbol{x}^\top \boldsymbol{U}_r^\top S_k S_k^\top \boldsymbol{U}_r \boldsymbol{x} = 0$. Since $S_k^\top \boldsymbol{U}_r \in \mathbb{R}^{k \times r}$ is distributed as an i.i.d Gaussian sketching matrix, we conclude that $\text{rank}(\boldsymbol{U}_r^\top S_k S_k^\top \boldsymbol{U}_r) = \text{rank}(S_k^\top \boldsymbol{U}_r) = r$ almost surely with $k \geq r$. Now with $S_k^\top \boldsymbol{\Sigma}_{p-r} S_k$ invertible and $\text{rank}(\boldsymbol{U}_r^\top S_k S_k^\top \boldsymbol{U}_r) = r$ (which happens almost surely), we know that $\boldsymbol{x}^\top \boldsymbol{U}_r^\top \boldsymbol{H} \boldsymbol{U}_r \boldsymbol{x} = 0$ iff $\boldsymbol{x} = 0$, or equivalently, $\boldsymbol{U}_r^\top \boldsymbol{H} \boldsymbol{U}_r$ is invertible.

Now we can safely write $(\boldsymbol{U}_r^\top \boldsymbol{H} \boldsymbol{U}_r)^{-1}$. In this case,

$$\boldsymbol{\lambda}^* = (\boldsymbol{U}_r^\top \boldsymbol{H} \boldsymbol{U}_r)^{-1} (\boldsymbol{U}_r^\top \boldsymbol{H} \boldsymbol{\Sigma}_{p-r} - \boldsymbol{U}_r^\top) \boldsymbol{\beta}. \tag{27}$$

Based on (26) and (27), we have

$$
\begin{aligned}
S_k \boldsymbol{\xi}^* &= \boldsymbol{H} \left( \boldsymbol{\Sigma}_{p-r} \boldsymbol{\beta} - \boldsymbol{U}_r (\boldsymbol{U}_r^\top \boldsymbol{H} \boldsymbol{U}_r)^{-1} (\boldsymbol{U}_r^\top \boldsymbol{H} \boldsymbol{\Sigma}_{p-r} - \boldsymbol{U}_r^\top) \boldsymbol{\beta} \right) \\
&= \boldsymbol{H} \left( \boldsymbol{\Sigma}_{p-r} - \boldsymbol{U}_r (\boldsymbol{U}_r^\top \boldsymbol{H} \boldsymbol{U}_r)^{-1} \boldsymbol{U}_r^\top (\boldsymbol{H} \boldsymbol{\Sigma}_{p-r} - \mathbf{I}) \right) \boldsymbol{\beta}.
\end{aligned}
$$

With the above expression at hand, we are ready to control quantity $(\boldsymbol{\beta} - S_k \boldsymbol{\xi}^*)^\top \boldsymbol{\Sigma}_{p-r} (\boldsymbol{\beta} - S_k \boldsymbol{\xi}^*)$. For the sake of notational simplicity, let us write

$$\boldsymbol{G} := \boldsymbol{U}_r (\boldsymbol{U}_r^\top \boldsymbol{H} \boldsymbol{U}_r)^{-1} \boldsymbol{U}_r^\top.$$

Then we obtain the following equalities:

$$
\begin{aligned}
S_k \boldsymbol{\xi}^* &= \boldsymbol{H} \left( \boldsymbol{\Sigma}_{p-r} - \boldsymbol{G} (\boldsymbol{H} \boldsymbol{\Sigma}_{p-r} - \mathbf{I}) \right) \boldsymbol{\beta}; \\
\boldsymbol{\beta} - S_k \boldsymbol{\xi}^* &= (\mathbf{I} - \boldsymbol{H} \boldsymbol{G})(\mathbf{I} - \boldsymbol{H} \boldsymbol{\Sigma}_{p-r}) \boldsymbol{\beta}.
\end{aligned}
$$

Putting the pieces together yields

$$(\boldsymbol{\beta} - S_k \boldsymbol{\xi}^*)^\top \boldsymbol{\Sigma}_{p-r} (\boldsymbol{\beta} - S_k \boldsymbol{\xi}^*) = \boldsymbol{\beta}^\top (\mathbf{I} - \boldsymbol{\Sigma}_{p-r} \boldsymbol{H})(\mathbf{I} - \boldsymbol{G} \boldsymbol{H}) \boldsymbol{\Sigma}_{p-r} (\mathbf{I} - \boldsymbol{H} \boldsymbol{G})(\mathbf{I} - \boldsymbol{H} \boldsymbol{\Sigma}_{p-r}) \boldsymbol{\beta}.$$

**Step 2: Upper bounding the minimal value.**

By recalling the notation $\boldsymbol{H} = S_k(S_k^\top \boldsymbol{\Sigma}_{p-r} S_k)^{-1} S_k^\top$, we know $\boldsymbol{H} \boldsymbol{\Sigma}_{p-r} \boldsymbol{H} = \boldsymbol{H}$ and $\boldsymbol{G} \boldsymbol{H} \boldsymbol{G} = \boldsymbol{G}$. Then $\boldsymbol{H} \boldsymbol{\Sigma}_{p-r} (\mathbf{I} - \boldsymbol{H} \boldsymbol{\Sigma}_{p-r}) = \mathbf{0}$, and

$$
\begin{aligned}
(\boldsymbol{\beta} - S_k \boldsymbol{\xi}^*)^\top \boldsymbol{\Sigma}_{p-r} (\boldsymbol{\beta} - S_k \boldsymbol{\xi}^*) &= \boldsymbol{\beta}^\top (\mathbf{I} - \boldsymbol{\Sigma}_{p-r} \boldsymbol{H})(\boldsymbol{\Sigma}_{p-r} - \boldsymbol{G} \boldsymbol{H} \boldsymbol{\Sigma}_{p-r} - \boldsymbol{\Sigma}_{p-r} \boldsymbol{H} \boldsymbol{G} + \boldsymbol{G})(\mathbf{I} - \boldsymbol{H} \boldsymbol{\Sigma}_{p-r}) \boldsymbol{\beta} \\
&= \boldsymbol{\beta}^\top (\mathbf{I} - \boldsymbol{\Sigma}_{p-r} \boldsymbol{H})(\boldsymbol{\Sigma}_{p-r} + \boldsymbol{G})(\mathbf{I} - \boldsymbol{H} \boldsymbol{\Sigma}_{p-r}) \boldsymbol{\beta} \\
&= \boldsymbol{\beta}^\top \boldsymbol{\Sigma}_{p-r} \boldsymbol{\beta} - \boldsymbol{\beta}^\top \boldsymbol{\Sigma}_{p-r} \boldsymbol{H} \boldsymbol{\Sigma}_{p-r} \boldsymbol{\beta} + \boldsymbol{\beta}^\top (\mathbf{I} - \boldsymbol{\Sigma}_{p-r} \boldsymbol{H}) \boldsymbol{G} (\mathbf{I} - \boldsymbol{H} \boldsymbol{\Sigma}_{p-r}) \boldsymbol{\beta} \\
&\leq \boldsymbol{\beta}^\top \boldsymbol{\Sigma}_{p-r} \boldsymbol{\beta} + \|(\boldsymbol{U}_r^\top \boldsymbol{H} \boldsymbol{U}_r)^{-1}\|_2 \|\boldsymbol{U}_r^\top \boldsymbol{\beta} - \boldsymbol{U}_r^\top \boldsymbol{H} \boldsymbol{\Sigma}_{p-r} \boldsymbol{\beta}\|_2^2.
\end{aligned}
$$

311    By definition of $\widetilde{S}_1 := \boldsymbol{U}_r^\top S_k$ and $\widetilde{S}_2 := \boldsymbol{U}_{p-r}^\top S_k$, we can see $\widetilde{S}_1$ and $\widetilde{S}_2$ are independent, and their

312    entries are independent standard Gaussian random variables. Additionally denoting $\widetilde{\boldsymbol{\beta}}_1 := \boldsymbol{U}_r^\top \boldsymbol{\beta}$ and

313    $\widetilde{\boldsymbol{\beta}}_2 := \boldsymbol{U}_{p-r}^\top \boldsymbol{\beta}$, we can rewrite the above as

$$(\boldsymbol{\beta}-S_k\boldsymbol{\xi}^*)^\top \boldsymbol{\Sigma}_{p-r}(\boldsymbol{\beta}-S_k\boldsymbol{\xi}^*) \leq \widetilde{\boldsymbol{\beta}}_2^\top \boldsymbol{\Lambda}_{p-r}\widetilde{\boldsymbol{\beta}}_2 + \|(\boldsymbol{U}_r^\top \boldsymbol{H}\boldsymbol{U}_r)^{-1}\|_2\|\widetilde{\boldsymbol{\beta}}_1 - \widetilde{S}_1(\widetilde{S}_2^\top \boldsymbol{\Lambda}_{p-r}\widetilde{S}_2)^{-1}\widetilde{S}_2^\top \boldsymbol{\Lambda}_{p-r}\widetilde{\boldsymbol{\beta}}_2\|_2^2.$$
(28)

314    With some algebra (see the details at the end of this section), it can be shown that

$$\|\widetilde{\boldsymbol{\beta}}_1 - \widetilde{S}_1(\widetilde{S}_2^\top \boldsymbol{\Lambda}_{p-r}\widetilde{S}_2)^{-1}\widetilde{S}_2^\top \boldsymbol{\Lambda}_{p-r}\widetilde{\boldsymbol{\beta}}_2\|_2^2 \leq 2\|\widetilde{\boldsymbol{\beta}}_1\|_2^2 + 2\|\widetilde{S}_1\|_2^2\|(\widetilde{S}_2^\top \boldsymbol{\Lambda}_{p-r}\widetilde{S}_2)^{-1}\|_2;$$
(29)

$$\|(\boldsymbol{U}_r^\top \boldsymbol{H}\boldsymbol{U}_r)^{-1}\|_2 \leq \frac{\lambda_{\max}(\widetilde{S}_2^\top \boldsymbol{\Lambda}_{p-r}\widetilde{S}_2)}{\lambda_{\min}(\widetilde{S}_1\widetilde{S}_1^\top)}.$$
(30)

315    Plugging inequalities (29) and (30) into (28) yields

$$(\boldsymbol{\beta} - S_k\boldsymbol{\xi}^*)^\top \boldsymbol{\Sigma}_{p-r}(\boldsymbol{\beta} - S_k\boldsymbol{\xi}^*)$$

$$\overset{\text{by (29)}}{\leq} 2\|(\boldsymbol{U}_r^\top \boldsymbol{H}\boldsymbol{U}_r)^{-1}\|_2\|\widetilde{\boldsymbol{\beta}}_1\|_2^2 + \left(1 + 2\|(\boldsymbol{U}_r^\top \boldsymbol{H}\boldsymbol{U}_r)^{-1}\|_2\|\widetilde{S}_1\|_2^2\|(\widetilde{S}_2^\top \boldsymbol{\Lambda}_{p-r}\widetilde{S}_2)^{-1}\|_2\right) \cdot \widetilde{\boldsymbol{\beta}}_2^\top \boldsymbol{\Lambda}_{p-r}\widetilde{\boldsymbol{\beta}}_2$$

$$\overset{\text{by (30)}}{\leq} 2\frac{\|\widetilde{S}_2^\top \boldsymbol{\Lambda}_{p-r}\widetilde{S}_2\|_2}{\lambda_{\min}(\widetilde{S}_1\widetilde{S}_1^\top)}\|\widetilde{\boldsymbol{\beta}}_1\|_2^2 + \left(1 + 2\frac{\lambda_{\max}(\widetilde{S}_2^\top \boldsymbol{\Lambda}_{p-r}\widetilde{S}_2)}{\lambda_{\min}(\widetilde{S}_2^\top \boldsymbol{\Lambda}_{p-r}\widetilde{S}_2)} \cdot \frac{\lambda_{\max}(\widetilde{S}_1\widetilde{S}_1^\top)}{\lambda_{\min}(\widetilde{S}_1\widetilde{S}_1^\top)}\right) \cdot \widetilde{\boldsymbol{\beta}}_2^\top \boldsymbol{\Lambda}_{p-r}\widetilde{\boldsymbol{\beta}}_2$$

$$= 2\frac{\|\widetilde{S}_2^\top \boldsymbol{\Lambda}_{p-r}\widetilde{S}_2\|_2}{\lambda_{\min}(\widetilde{S}_1\widetilde{S}_1^\top)}\|\widetilde{\boldsymbol{\beta}}_1\|_2^2 + \left(1 + 2\kappa(\widetilde{S}_2^\top \boldsymbol{\Lambda}_{p-r}\widetilde{S}_2)\kappa(\widetilde{S}_1\widetilde{S}_1^\top)\right) \cdot \widetilde{\boldsymbol{\beta}}_2^\top \boldsymbol{\Lambda}_{p-r}\widetilde{\boldsymbol{\beta}}_2.$$

316    This completes the proof of Lemma C.5.

317    **Proof of (29) and (30).**    First we show (29). By the triangle inequality, we have

$$\|\widetilde{\boldsymbol{\beta}}_1 - \widetilde{S}_1(\widetilde{S}_2^\top \boldsymbol{\Lambda}_{p-r}\widetilde{S}_2)^{-1}\widetilde{S}_2^\top \boldsymbol{\Lambda}_{p-r}\widetilde{\boldsymbol{\beta}}_2\|_2^2 \leq 2\|\widetilde{\boldsymbol{\beta}}_1\|_2^2 + 2\|\widetilde{S}_1(\widetilde{S}_2^\top \boldsymbol{\Lambda}_{p-r}\widetilde{S}_2)^{-1}\widetilde{S}_2^\top \boldsymbol{\Lambda}_{p-r}\widetilde{\boldsymbol{\beta}}_2\|_2^2.$$

318    Note that for $\boldsymbol{A} \in \mathbb{R}^{p \times p}$ and $\boldsymbol{x} \in \mathbb{R}^p$, the multiplicative property of the norm shows $\|\boldsymbol{A}\boldsymbol{x}\|_2 \leq$

319    $\|\boldsymbol{A}\|_2\|\boldsymbol{x}\|_2$. Using this property, it can be seen that

$$\|\widetilde{S}_1(\widetilde{S}_2^\top \boldsymbol{\Lambda}_{p-r}\widetilde{S}_2)^{-1}\widetilde{S}_2^\top \boldsymbol{\Lambda}_{p-r}\widetilde{\boldsymbol{\beta}}_2\|_2^2 \leq \|\widetilde{S}_1\|_2^2\|(\widetilde{S}_2^\top \boldsymbol{\Lambda}_{p-r}\widetilde{S}_2)^{-1}\widetilde{S}_2^\top \boldsymbol{\Lambda}_{p-r}^{1/2}\|_2^2\|\boldsymbol{\Lambda}_{p-r}^{1/2}\widetilde{\boldsymbol{\beta}}_2\|_2^2.$$

320    By $\|\boldsymbol{A}\boldsymbol{A}^\top\|_2 = \|\boldsymbol{A}\|_2^2$, it follows that

$$\|(\widetilde{S}_2^\top \boldsymbol{\Lambda}_{p-r}\widetilde{S}_2)^{-1}\widetilde{S}_2^\top \boldsymbol{\Lambda}_{p-r}^{1/2}\|_2^2 = \|(\widetilde{S}_2^\top \boldsymbol{\Lambda}_{p-r}\widetilde{S}_2)^{-1}\|_2.$$

321    Then we have

$$\|\widetilde{\boldsymbol{\beta}}_1 - \widetilde{S}_1(\widetilde{S}_2^\top \boldsymbol{\Lambda}_{p-r}\widetilde{S}_2)^{-1}\widetilde{S}_2^\top \boldsymbol{\Lambda}_{p-r}\widetilde{\boldsymbol{\beta}}_2\|_2^2 \leq 2\|\widetilde{\boldsymbol{\beta}}_1\|_2^2 + 2\|\widetilde{S}_1\|_2^2\|(\widetilde{S}_2^\top \boldsymbol{\Lambda}_{p-r}\widetilde{S}_2)^{-1}\|_2.$$

322    It remains to show (30). By definition, for a symmetric matrix $\boldsymbol{A}$, we can write $\lambda_{\min}(\boldsymbol{A}) =$

323    $\min_{\|\boldsymbol{x}\|_2=1} \boldsymbol{x}^\top \boldsymbol{A}\boldsymbol{x}$. Taking $\forall \boldsymbol{x} \in \mathbb{R}^r$ with $\|\boldsymbol{x}\|_2 = 1$, we have

$$\boldsymbol{x}^\top \boldsymbol{U}_r^\top \boldsymbol{H}\boldsymbol{U}_r\boldsymbol{x} = \boldsymbol{x}^\top \boldsymbol{U}_r^\top S_k(S_k^\top \boldsymbol{\Sigma}_{p-r}S_k)^{-1}S_k^\top \boldsymbol{U}_r\boldsymbol{x} = \boldsymbol{x}^\top \widetilde{S}_1(\widetilde{S}_2^\top \boldsymbol{\Lambda}_{p-r}\widetilde{S}_2)^{-1}\widetilde{S}_1^\top \boldsymbol{x}$$

$$\geq \lambda_{\min}((\widetilde{S}_2^\top \boldsymbol{\Lambda}_{p-r}\widetilde{S}_2)^{-1})(\boldsymbol{x}^\top \widetilde{S}_1\widetilde{S}_1^\top \boldsymbol{x})$$

$$\geq \lambda_{\min}((\widetilde{S}_2^\top \boldsymbol{\Lambda}_{p-r}\widetilde{S}_2)^{-1})\lambda_{\min}(\widetilde{S}_1\widetilde{S}_1^\top) = \frac{\lambda_{\min}(\widetilde{S}_1\widetilde{S}_1^\top)}{\lambda_{\max}(\widetilde{S}_2^\top \boldsymbol{\Lambda}_{p-r}\widetilde{S}_2)}.$$

324    Then we know

$$\|(\boldsymbol{U}_r^\top \boldsymbol{H}\boldsymbol{U}_r)^{-1}\|_2 = \frac{1}{\lambda_{\min}(\boldsymbol{U}_r^\top \boldsymbol{H}\boldsymbol{U}_r)} \leq \frac{\lambda_{\max}(\widetilde{S}_2^\top \boldsymbol{\Lambda}_{p-r}\widetilde{S}_2)}{\lambda_{\min}(\widetilde{S}_1\widetilde{S}_1^\top)}.$$

# D  Auxiliary proofs

## D.1  Proof of Lemma C.3

Let us write the singular value decomposition of $\boldsymbol{\Sigma}$ as $\boldsymbol{\Sigma} = \boldsymbol{U}\boldsymbol{\Lambda}\boldsymbol{U}^\top$ with $\boldsymbol{\Lambda} = \mathrm{diag}(\boldsymbol{\lambda})$, $\boldsymbol{\lambda} \in \mathbb{R}^p$. Then we have $\mathrm{tr}(\boldsymbol{\Sigma}) = \|\boldsymbol{\lambda}\|_1$, $\|\boldsymbol{\Sigma}\|_F = \|\boldsymbol{\lambda}\|_2$, $\mathrm{tr}(\boldsymbol{\Sigma}^2) = \|\boldsymbol{\lambda}\|_2^2$ and $\mathrm{tr}(\boldsymbol{\Sigma}^4) = \|\boldsymbol{\lambda}\|_4^4$. With the new notation, the claim of Lemma C.3 is now equivalent to $\|\boldsymbol{\lambda}\|_1^2\|\boldsymbol{\lambda}\|_4 \geq \|\boldsymbol{\lambda}\|_2^3$.

We prove $\|\boldsymbol{\lambda}\|_1^2\|\boldsymbol{\lambda}\|_4 \geq \|\boldsymbol{\lambda}\|_2^3$ using the following ingredients:

$$\text{(i) } \|\boldsymbol{\lambda}\|_3^3\|\boldsymbol{\lambda}\|_1 \geq \|\boldsymbol{\lambda}\|_2^4;$$

$$\text{(ii) } \|\boldsymbol{\lambda}\|_4^4\|\boldsymbol{\lambda}\|_1 \geq \|\boldsymbol{\lambda}\|_3^3\|\boldsymbol{\lambda}\|_2^2;$$

$$\text{(iii) } \|\boldsymbol{\lambda}\|_1 \geq \|\boldsymbol{\lambda}\|_2,$$

where (i) holds directly from Cauchy-Schwarz inequality; (ii) follows from the equality

$$\|\boldsymbol{\lambda}\|_4^4\|\boldsymbol{\lambda}\|_1 - \|\boldsymbol{\lambda}\|_3^3\|\boldsymbol{\lambda}\|_2^2 = \frac{1}{2}\sum_{i \neq j}\lambda_i\lambda_j(\lambda_i + \lambda_j)(\lambda_i - \lambda_j)^2 \geq 0$$

with $\lambda_i \geq 0$; (iii) follows from the observation that $\|\boldsymbol{\lambda}\|_1^2 - \|\boldsymbol{\lambda}\|_2^2 = \sum_{i \neq j}\lambda_i\lambda_j \geq 0$ with $\lambda_i \geq 0$.

Then we have

$$\|\boldsymbol{\lambda}\|_1^8\|\boldsymbol{\lambda}\|_4^4 = \left(\frac{\|\boldsymbol{\lambda}\|_1\|\boldsymbol{\lambda}\|_4^4}{\|\boldsymbol{\lambda}\|_3^3}\right) \cdot (\|\boldsymbol{\lambda}\|_1\|\boldsymbol{\lambda}\|_3^3) \cdot (\|\boldsymbol{\lambda}\|_1^6) \geq \|\boldsymbol{\lambda}\|_2^2 \cdot \|\boldsymbol{\lambda}\|_2^4 \cdot \|\boldsymbol{\lambda}\|_2^6 = \|\boldsymbol{\lambda}\|_2^{12}.$$

Thus we show $\|\boldsymbol{\lambda}\|_1^2\|\boldsymbol{\lambda}\|_4 \geq \|\boldsymbol{\lambda}\|_2^3$, and Lemma C.3 follows.

## D.2  Proof of Lemma C.7

We closely follow the proof of Theorem 5.39 in [8] that uses a covering argument with three steps: 1) discretization; 2) concentration; 3) union bound. In the discretization step, we discretize the problem using a net $\mathcal{N}$; in the concentration step, we bound $\|\boldsymbol{A}\boldsymbol{x}\|_2$ for each $\boldsymbol{x} \in \mathcal{N}$. Finally, we use the union bound to establish a concentration bound over $\boldsymbol{x} \in \mathcal{S}^{n-1}$.

**Step 1: Discretization.**  First we invoke Lemma 5.36 in [8]:

**Lemma D.1.**  *Consider a matrix $\boldsymbol{B}$ that satisfies*

$$\|\boldsymbol{B}^\top\boldsymbol{B} - \mathbf{I}\|_2 \leq \max(\delta, \delta^2)$$

*for some $\delta > 0$. Then*

$$1 - \delta \leq s_{\min}(\boldsymbol{B}) \leq s_{\max}(\boldsymbol{B}) \leq 1 + \delta.$$

*Conversely, if $\boldsymbol{B}$ satisfies $1 - \delta \leq s_{\min}(\boldsymbol{B}) \leq s_{\max}(\boldsymbol{B}) \leq 1 + \delta$ for some $\delta > 0$, then $\|\boldsymbol{B}^\top\boldsymbol{B} - \mathbf{I}\|_2 \leq 3\max(\delta, \delta^2)$.*

Write $T = \|\boldsymbol{\Lambda}\|_2^2$ and $\boldsymbol{A} = \boldsymbol{\Lambda}\boldsymbol{S}$. Then the claim is equivalent to

$$\left\|\frac{1}{T}\boldsymbol{A}^\top\boldsymbol{A} - \mathbf{I}\right\|_2 \leq \max(t, t^2) = t.$$

We can evaluate the operator norm on a $1/4$-net $\mathcal{N}$ of the unit sphere $\mathcal{S}^{n-1}$: with Lemma 5.4 in [8], we have

$$\left\|\frac{1}{T}\boldsymbol{A}^\top\boldsymbol{A} - \mathbf{I}\right\|_2 \leq 2\max_{\boldsymbol{x} \in \mathcal{N}}\left|\frac{1}{T}\|\boldsymbol{A}\boldsymbol{x}\|_2^2 - 1\right|.$$

Note that we can choose $\mathcal{N}$ such that $|\mathcal{N}| \leq 9^n$.

**Step 2: Concentration.** Fix $x \in \mathcal{S}^{n-1}$. Denote the $i$-th row of matrix $A$ and $S$ by $A_i$ and $S_i$, respectively. Then $\langle A_i, x \rangle / \lambda_i = \langle S_i, x \rangle \sim \mathcal{N}(0,1)$ and the $\langle A_i, x \rangle$'s are independent to each other. We can express $\|Ax\|_2^2$ as a sum of independent random variables

$$\|Ax\|_2^2 = \sum_{i=1}^{N} \langle A_i, x \rangle^2 =: \sum_{i=1}^{N} \lambda_i^2 Z_i^2,$$

where $Z_i \overset{iid}{\sim} \mathcal{N}(0,1)$. By Lemma 1 of [5], we have

$$P\left( \left| \frac{1}{\sum_{i=1}^{N} \lambda_i^2} \|Ax\|_2^2 - 1 \right| \geq 2 \frac{\sqrt{\sum_{i=1}^{N} \lambda_i^4}}{\sum_{i=1}^{N} \lambda_i^2} \sqrt{\delta} + 2 \frac{\max_{1 \leq i \leq N} \lambda_i^2}{\sum_{i=1}^{N} \lambda_i^2} \delta \right) \leq 2 e^{-\delta}.$$

When $\delta = \min\left\{ \frac{1}{16} \frac{\|\lambda\|_2^4}{\|\lambda\|_4^4} t^2, \frac{1}{4} \frac{\|\lambda\|_2^2}{\|\lambda\|_\infty^2} t \right\}$, we have $2 \frac{\sqrt{\sum_{i=1}^{N} \lambda_i^4}}{\sum_{i=1}^{N} \lambda_i^2} \sqrt{\delta} \leq \frac{1}{2} t$ and $2 \frac{\max_{1 \leq i \leq N} \lambda_i^2}{\sum_{i=1}^{N} \lambda_i^2} \delta \leq \frac{1}{2} t$. Then we can rewrite the tail bound as

$$P\left( \left| \frac{1}{\sum_{i=1}^{N} \lambda_i^2} \|Ax\|_2^2 - 1 \right| \geq t \right) \leq 2 \exp\left( -\min\left\{ \frac{1}{16} \frac{\|\lambda\|_2^4}{\|\lambda\|_4^4} t^2, \frac{1}{4} \frac{\|\lambda\|_2^2}{\|\lambda\|_\infty^2} t \right\} \right).$$

**Step 3: Union bound.** Taking the bound over all vectors in the net $\mathcal{N}$, we obtain

$$P\left( \max_{x \in \mathcal{N}} \left| \frac{1}{T} \|Ax\|_2^2 - 1 \right| \geq t \right) \leq 9^n \cdot 2 \exp\left( -\min\left\{ \frac{1}{16} \frac{\|\lambda\|_2^4}{\|\lambda\|_4^4} t^2, \frac{1}{4} \frac{\|\lambda\|_2^2}{\|\lambda\|_\infty^2} t \right\} \right).$$

Thus, by Lemma D.1, we have, for $t < 1$,

$$(1-t)\sqrt{\sum_{i=1}^{N} \lambda_i^2} \leq s_{\min}(A) \leq s_{\max}(A) \leq (1+t)\sqrt{\sum_{i=1}^{N} \lambda_i^2}$$

with probability at least $1 - 9^n \cdot 2 \exp\left( -\min\left\{ \frac{1}{16} \frac{\|\lambda\|_2^4}{\|\lambda\|_4^4} t^2, \frac{1}{4} \frac{\|\lambda\|_2^2}{\|\lambda\|_\infty^2} t \right\} \right)$.

### D.3 Technical details of Theorem A.1

In this part we check some technical details of Theorem A.1. Recall from the proof of Theorem 1 that the sketched linear model is

$$y_i = \langle \widetilde{x}_i, \beta^S \rangle + z_i^S = \langle S_k^\top x_i, \beta^S \rangle + z_i^S,$$

where $z_i^S = \langle x_i, \beta \rangle + \sigma z_i - \langle \widetilde{x}_i, \beta^S \rangle$ and $\beta^S = (S_k^\top \Sigma S_k)^{-1} S_k^\top \Sigma \beta$. We are essentially testing whether sketched coefficients $\beta^S$ are zero or not as

$$H_0^S : \beta^S = 0 \quad \text{versus} \quad H_1^S : \beta^S \neq 0.$$

In what follows, we verify that the technical conditions of Theorem 2.1 and Corollary 2.2 in [7] are satisfied under assumptions **(B1, B2)** and the sketched model $y_i = \langle \widetilde{x}_i, \beta^S \rangle + z_i^S$. This verification step directly leads to the desired result in Theorem A.1. See Section 2.1 of [7] for the technical conditions; specifically, it suffices to verify **(A1)***(a,b,c,d)* and **(A2)** therein. We write them as **(S-A1)***(a,b,c,d)* and **(S-A2)** below.

**Verification of (S-A1):** By our assumption **(B1)** with $\widetilde{x}_i = S_k^\top \Gamma u_i$, we can directly see assumptions **(S-A1)***(a,b,c,d)* are satisfied.

**Verification of (S-A2):** It suffices to check the following two conditions:

$$\mathbb{E}\left[ \left( \mathbb{E}\left[ (z_i^S)^4 | \widetilde{x}_i \right] \right)^2 \right] = O(1) \quad \text{and} \quad \max_{i=1}^{n} \mathbb{E}\left[ (z_i^S)^4 | \widetilde{x}_i \right] = o_P(\sqrt{k}). \tag{31}$$

**First claim of** (31). To simplify notation, write $\boldsymbol{\delta} := \boldsymbol{\beta} - S_k\boldsymbol{\beta}^S$. Then we can write

$$z_i^S = \sigma z_i + \boldsymbol{\delta}' \boldsymbol{x}_i.$$

We first derive the expression for $\mathbb{E}[(z_i^S)^4|\widetilde{\boldsymbol{x}}_i]$. Notice that $\mathbb{E}[(z_i^S)^4|\widetilde{\boldsymbol{x}}_i] = \mathbb{E}[\mathbb{E}[(z_i^S)^4|\boldsymbol{x}_i] \mid \widetilde{\boldsymbol{x}}_i]$, with

$$\mathbb{E}\big[\big(z_i^S\big)^4 \,|\boldsymbol{x}_i\big] = \mathbb{E}\left[(\sigma z_i + \boldsymbol{\delta}'\boldsymbol{x}_i)^4 \,|\boldsymbol{x}_i\right] \leq 8c\sigma^4 + 8\left(\boldsymbol{\delta}'\boldsymbol{x}_i\right)^4. \tag{32}$$

The above inequality follows by $(x+y)^4 \leq 8(x^4 + y^4)$ as well as assumption **(B2)**. Then we further have

$$\mathbb{E}\left[\left(\mathbb{E}\left[(z_i^S)^4|\widetilde{\boldsymbol{x}}_i\right]\right)^2\right] = \mathbb{E}\left[\left(8c\sigma^4 + 8\mathbb{E}\left[(\boldsymbol{\delta}'\boldsymbol{x}_i)^4|\widetilde{\boldsymbol{x}}_i\right]\right)^2\right] \leq 128\left(c^2\sigma^8 + \mathbb{E}\left[\left(\mathbb{E}\left[(\boldsymbol{\delta}'\boldsymbol{x}_i)^4|\widetilde{\boldsymbol{x}}_i\right]\right)^2\right]\right). \tag{33}$$

To show the first claim in (31), it suffices to show $\mathbb{E}[(\mathbb{E}[(\boldsymbol{\delta}'\boldsymbol{x}_i)^4|\widetilde{\boldsymbol{x}}_i])^2] = O(1)$. By $\mathrm{Var}(\mathbb{E}[Y|X]) \leq \mathrm{Var}(Y)$, we have

$$\mathbb{E}\left[\left(\mathbb{E}\left[(\boldsymbol{\delta}'\boldsymbol{x}_i)^4|\widetilde{\boldsymbol{x}}_i\right]\right)^2\right] = \mathrm{Var}\left(\mathbb{E}\left[(\boldsymbol{\delta}'\boldsymbol{x}_i)^4|\widetilde{\boldsymbol{x}}_i\right]\right) + \left(\mathbb{E}\left[\mathbb{E}\left[(\boldsymbol{\delta}'\boldsymbol{x}_i)^4|\widetilde{\boldsymbol{x}}_i\right]\right]\right)^2$$

$$\leq \mathrm{Var}\left((\boldsymbol{\delta}'\boldsymbol{x}_i)^4\right) + \left(\mathbb{E}\left[(\boldsymbol{\delta}'\boldsymbol{x}_i)^4\right]\right)^2 = \mathbb{E}\left[(\boldsymbol{\delta}'\boldsymbol{x}_i)^8\right].$$

With $\boldsymbol{\delta} = \boldsymbol{\beta} - S_k\boldsymbol{\beta}^S$, we also have

$$\mathbb{E}\left[(\boldsymbol{\delta}'\boldsymbol{x}_i)^8\right] = \mathbb{E}\left[\langle \boldsymbol{x}_i, \boldsymbol{\beta} - S_k\boldsymbol{\beta}^S\rangle^8\right] \leq \|\boldsymbol{\Gamma}^\top(\boldsymbol{\beta} - S_k\boldsymbol{\beta}^S)\|_2^8 \sup_{\|v\|_2=1}(\mathbb{E}|v'\boldsymbol{u}_i|^8). \tag{34}$$

By definition of $\boldsymbol{\beta}^S$, we know $\|\boldsymbol{\Gamma}^\top(\boldsymbol{\beta} - S_k\boldsymbol{\beta}^S)\|_2^2 = \boldsymbol{\beta}^\top\boldsymbol{\Sigma}\boldsymbol{\beta} - \Delta_k^2 \leq \boldsymbol{\beta}^\top\boldsymbol{\Sigma}\boldsymbol{\beta} = o(1)$. By **(B1)**(b), we further know $\sup_{\|v\|=1}(\mathbb{E}|v'\boldsymbol{u}_i|^8) = O(1)$. Thus we show $\mathbb{E}[(\mathbb{E}[(\boldsymbol{\delta}'\boldsymbol{x}_i)^4|\widetilde{\boldsymbol{x}}_i])^2] = O(1)$. Therefore, together with inequality (33), the first claim in (31) follows.

**Second claim of** (31). Next we show the second claim in (31). By inequality (32), we have

$$\max_{i=1}^n \mathbb{E}\left[(z_i^S)^4|\widetilde{\boldsymbol{x}}_i\right] \leq 8c\sigma^4 + 8\max_{i=1}^n \mathbb{E}\left[(\boldsymbol{\delta}'\boldsymbol{x}_i)^4|\widetilde{\boldsymbol{x}}_i\right],$$

and it suffices to show that $\max_{i=1}^n \mathbb{E}\left[(\boldsymbol{\delta}'\boldsymbol{x}_i)^4|\widetilde{\boldsymbol{x}}_i\right] = o_P(\sqrt{k})$. Observe that

$$\mathbb{P}\left(\max_{i=1}^n \mathbb{E}\left[(\boldsymbol{\delta}'\boldsymbol{x}_i)^4|\widetilde{\boldsymbol{x}}_i\right] \geq \epsilon\right) \overset{(i)}{\leq} \frac{n}{\epsilon^2}\mathrm{Var}\left((\boldsymbol{\delta}'\boldsymbol{x}_i)^4\right) \overset{(ii)}{\leq} n\boldsymbol{\beta}^\top\boldsymbol{\Sigma}\boldsymbol{\beta} \frac{\sup_{\|v\|=1}(\mathbb{E}|v'\boldsymbol{u}_i|^8)}{\epsilon^2} \overset{(iii)}{=} o\left(\frac{k}{\epsilon^2}\right).$$

In the above argument, step (i) follows from the union bound and Chebyshev's inequality; step (ii) is from (34) and $\mathrm{Var}\left((\boldsymbol{\delta}'\boldsymbol{x}_i)^4\right) \leq \mathbb{E}[(\boldsymbol{\delta}'\boldsymbol{x}_i)^8]$; step (iii) uses the local alternative $\boldsymbol{\beta}^\top\boldsymbol{\Sigma}\boldsymbol{\beta} = o(k/n)$ and assumption **(B1)**(b). Therefore we can conclude that $\max_{i=1}^n \mathbb{E}[(\boldsymbol{\delta}'\boldsymbol{x}_i)^4|\widetilde{\boldsymbol{x}}_i] = o_P(\sqrt{k})$, which completes the proof.