[Reviews · NeurIPS 2020]

Review 1

Summary and Contributions: Concerning the classical high-dimensional linear regression model Y=X\beta+Z, the submission considers testing H_0: \beta = 0 using a combination of random projection and the classic F test. The idea is quite natural yet proven to be optimal to some extent.

Strengths: Overall, this is a good submission to me; the studied problem is important, the proposed solution is solid, the justification is meaningful, and the authors managed to produce some neat and self-complete results in quite limited a given space.

Weaknesses: My questions are minor. (1) I did not find explicit assumptions about the noise distribution in the main text. It appears to be put in the supplement, which is not easy to locate. (2) Section 3 is difficult to follow. The theorem therein, for example, was stated in an informal way; the authors did not characterize the H_0 and H_1 probability space upon which the optimality results are derived. Are we considering the case where X is multivariate Gaussian, Z is standard Gaussian, and X is independent of Z? (3) Proposition 1 has to be developed with more efforts put in justifying Assumption (A). The current version makes it a little bit Bayesian as obviously the parameters were assumed to be random there. Shall we stick to this Bayesian type model throughout the whole paper?

Correctness: They read meaningful to me.

Clarity: Overall, this is a very well written paper. However, the references contain many unnecessary "et al"; please delete them all.

Relation to Prior Work: I believe so.

Reproducibility: Yes

Additional Feedback: It would be an interesting future topic to discuss the applicability/optimality of the proposed random projection idea to the type of problems studied in "Minimax rate of testing in sparse linear regression" which the authors have heavily cited.


Review 2

Summary and Contributions: This paper considers the problem of testing the global null hypothesis on the parameters of a high dimensional linear regression model. This is the equivalent of an F-test in standard, low dimensional settings. The idea proposed in the paper is pretty intuitive and amounts to sketching the design matrix, and then performing a standard F-test. Results are provided about the power of the method, and comparisons with related work are drawn with synthetic data.

Strengths: This paper is well written, the proposed method is simple and intuitive, and the potential benefits are clearly described and illustrated through simulations.

Weaknesses: Major comments: 1) One key concern with the method, and, apparently, with other methods in related literature, is that the power of the test deteriorates rapidly for larger \sigma. While this is not a problem, in general, with the standard F-test in dimensional problems, it may be an important problem in high dimensional settings where we expect low signal-to-noise ratios. In the experiments of Section 4, for instance, the power drops dramatically if, say, we set \sigma=10. 2) On a related note, the experimental section does not make a particularly strong case. The performance of tests depends on several factors: \sigma, distribution of X, strength of \beta, error distribution, p/n ratio, etc. But none of these factors are being investigated. I am also particularly surprised that current methods (ZC and CGZ in the paper --- I believe there is a typo in Table 2) are doing so poorly in this simple setting (their size is severely distorted). One way to address this concern would be perhaps to describe cases/settings where the proposed test is expected to perform poorly, so that the reader has a clearer picture of strengths/limitations. Minor comments: 1) I had an overarching concern that the paper may not be 'sufficiently novel'. Using an F-test after sketching is pretty intuitive and straightforward. I don't mean to de-value the contribution in the paper. Indeed, it is great to know that sketching works for this problem, but I would expect some modified F-test to address low signal-to-noise regimes. 2) It is confusing throughout the paper what is the distribution of z_i. I believe in the main analysis it is assumed to be normal but there may be extensions to non-normal settings? It is definitely normal/t_2 in the experiments. This should be made clear early, after the equation in Line 37. 3) Lemma 1 is well known. The "local alternative" condition on \beta^T \Sigma \beta simplifies the calculation of power, but it makes a small difference in deriving the power of the F-test, as it is well known from standard textbooks (e.g, "Multivariate Analysis" of Mardia et al.) 4) Line 189: "From the definition, it is clear that...". I don't think it is clear at all. Especially since \Delta_k is random. 5) Line 195: "Assumption (A) is standard when there is no prior information available for the alternatives". I don't think this statement makes much sense. It seems to be making a Bayesian argument about prior distribution on \beta, which contradicts the frequentist approach in the paper.

Correctness: Yes.

Clarity: The paper is well written.

Relation to Prior Work: Related work is adequately discussed.

Reproducibility: Yes

Additional Feedback:


Review 3

Summary and Contributions: The paper proposes a sketched F-test with desirable statistical properties in the linear regression regime where the number of features grows porportionally to the number of samples. Power analyses provide theoretical guarantees and algorithms are provided to efficently to test the statistic in practice.

Strengths: The paper addresses an interesting problem in directly addressing the cases in linear hypothesis testing where the number of features is close to the number of samples. This is a particularly important problem in a number of application domains where large scale data collection is difficult but for each sample a reasonable amount of features can be measured. The sketching angle of attack is natural and leads to both nice analysis and algorithms. It is clear that this technique could be broadly applied to other testing setups in the porportional n/p regime. The proposed procedure is extremely simple to implement and understand.

Weaknesses: The current presentation only addresses the hypothesis testing setup where one is interested in identifying where any of the features have a nonzero coefficient. For the specific results in this paper have limited application value. The authors address this as a natural progression for future work, but in its current form the contributions here may only be valuable towards future theoretical results. The experimental evaluation could be more elaborate. The specific simulations conducted suggest the method does indeed work given the assumptions presented, it's not immediately clear how well they play out in practice (where eigen spectra may not behave as nicely as in simulation, among other assumptions that may not be completely satisfied.

Correctness: I have not gone through the proofs in detail. The code provided is sufficient to replicate all simulation studies in the paper.

Clarity: The paper is very clearly written and easy to follow. Someone with a minimal background in linear regression hypothesis testing and a basic understanding of asymptotic analysis should have no problem understanding the paper.

Relation to Prior Work: The paper does a good job in provided strong motivation for the contributions by contrasting and comparing existing literature. The analysis is clearly novel, given most existing work focuses on specific assumptions (low intrinsic dimension, structured assumptions) or different asymptotic regimes.

Reproducibility: Yes

Additional Feedback: Table 2 caption should be just Type 2 error rates correct? The claim that Type 1 error is controlled "reasonably well" feels weak, if the results could be included (in main or appendix) it may strengthen the argument. While the paper's broad impact is somewhat limited, the contribution is a strong step towards tight analysis for porportional n/p regimes in many different hypothesis testing setups, and the sketching procedure seems to be more widely applicable. [Post-Rebuttal]: The authors effectively addressed the few concerns raised in reviews, and demonstrate that their contribution can easily be extended to other testing setups. With minor changes as discussed during the review and rebuttal this is a strong NeurIPS submission.


Review 4

Summary and Contributions: This paper presents a random projection method for performing an F-test of the null hypothesis in a linear model that all coefficients are equal to zero. The method is simple to compute and only requires one parameter (projection dimension). Theoretical results characterize the asymptotic power in the high-dimensional setting when the number of variables and observations grow proportionally and with few assumptions on the model. Moreover, this test is shown to be asymptotically more powerful than existing approaches under very general conditions, and it is minimax optimal when the model for the data is low-dimensional. Simulation results validate the effectiveness of this approach.

Strengths: This work studies a problem relevant to the community (testing in high dimensional regression) and shows how a simple and computationally efficient method (random sketching) can be deployed in this setting. The paper presents theoretical guarantees on its asymptotic power, and show that this method is statistically more efficient (both in theory and simulations) with respect to other existing approaches with theoretical guarantees. While the random sketching methodology is not new, the results provided in this paper can be used as a theoretical ground for developing other methodologies for testing in high dimensional data.

Weaknesses: One limitation I see of the results is that the global null is not as practically interesting as testing the significance of individual parameters, or a subset of parameters. The authors discuss at the end of the paper that this procedure can be extended to this scenario but it is not clear to me how, so it might be worth that the authors are more explicit in this extension. ------------------------------------------------------- Update: I would like to thank the authors for answering my comments in the rebuttal letter. The observations and additional simulations included clarified my questions about extensions of the method and the significance of the theoretical results.

Correctness: The claims and results in the paper seem to be correct, and are supported by simulation experiments, which also show some of the advantages of their approach.

Clarity: The paper is clearly written. For the sake of the presentation, the authors presented a simplified version of the results on the main paper, and defer the more general versions to the appendix, which I find it makes the presentation of the paper better.

Relation to Prior Work: The authors discuss in the introduction how their method requires fewer assumptions for the regime in which p and n are proportional than most of the existing high-dimensional regression tests. The authors are more specific in their discussion relating their method to [47] and [15], and provide results on ARE as well as empirical simulation comparisons. In terms of their methodology, the authors discuss other random sketching methods, and also mention the relationship of their assumptions with the work of [29]. However, I wonder whether the authors can relate the novelty of their theory and the proof techniques used here to other methods in the random sketching literature.

Reproducibility: Yes

Additional Feedback: The test discussed by the authors only considers the null hypothesis that all coefficients are equal to zero, whereas [47,15] consider a slightly more general null in which the coefficients are equal to some pre-specified value beta_0. I wonder whether the results of this paper can also consider this case. While the authors discuss the computational advantages of their methodology, I wonder whether the authors have some empirical evaluations of this.

[Author Response · NeurIPS 2020]

We thank all reviewers for very helpful comments. This letter addresses the major questions raised by the reviewers.

**Specific questions by Reviewer 1.** Please see the response below for "distribution assumptions" and "global null and group of coefficients". We will correct our references and typos in the table. Thanks.

**Specific questions by Reviewer 2.** (1) We agree with the reviewer that the low signal-to-noise ratio (SNR) setting is very interesting and hurts us in high dimensions. Indeed, when the SNR is below a certain threshold, *no* testing procedure can succeed; therefore our goal is to completely characterize this detection boundary and establish a valid test as soon as the SNR exceeds this detection boundary. Our upper and lower bounds established in Section 3 are steps towards such exact characterizations. (2) We refer the reviewer to our response below for "more discussions of our simulations". On the design of the simulations, in the first part of the simulation section, we have varied the following factors: ● signal-to-noise ratio ● distribution of $X$ ● distribution of noise by varying (i) the Frobenius norm of $\Sigma$; (ii) the eigen-structure of $\Sigma$; (iii) the noise distribution for different scenarios. We also examine the effect of dimension in the second part of the simulation section. We shall elaborate more in our revised version to make these more clear.

Other comments: (1) While the optimality of our proposed test is validated in various settings, the optimal procedure in the most general form still remains open. We agree it is very interesting and non-trivial to see whether a modified $F$-test can be adapted for optimal testing in the low-SNR setting. (2) We share the reviewer's intuition that Lemma 1 should hold as in the small $p$ case. However, it is non-trivial to demonstrate this result in the proportional regime where $p$ grows together with $n$. In addition, considering the local alternatives for small $\beta^T \Sigma \beta$, only makes the testing problems more challenging. (3) The ARE compares the samples required for two tests to achieve the same power and it is defined as $n_2/n_1$. Therefore, $\mathrm{ARE} < 1$ means the first test is more effective. Sorry there was a typo in our manuscript and we will correct it in the revised version. Thanks for catching this. Indeed, there is randomness associated with $S_k$ in the testing procedure. To conquer this issue, we provided high-probability guarantees in Section 2 and 3. Please also see the response below for "distribution assumptions".

**Specific questions by Reviewer 3.** Please see below for "global null and group of coefficients" and "more discussions of our simulations". We will elaborate more in our revised version when the eigen-spetra are not as nicely behaved. For Table 2, the Type I error rates are reflected in the column $\|\beta\|_2 = 0$; we shall make it more clear in the revised version.

**Specific questions by Reviewer 4.** Please see the response below for "global null and group of coefficients" and "novelty of the method and theory". We will add the comparisons of the running time in our revision. Thanks.

**Global null and group of coefficients.** To better demonstrate our main idea, our focus so far is mainly on testing the global null, however, our results and techniques can be substantially extended to testing other hypotheses. Built upon an improved argument of the high-dimensional $F$-test (see [36]), our framework can be *provably* adapted to testing whether $H_0 : G\boldsymbol{\beta} = r_0$ or $H_1 : G\boldsymbol{\beta} \neq r_0$ for $G \in \mathbb{R}^{q \times p}$ and $r_0 \in \mathbb{R}^q$ with $q \leq p$. For example, to test the joint significance of a group of coefficients, our test combines a sketching step (over the complement set of features) with the classical $F$-test. Fig 1 demonstrates its efficacy in the same setting as in our Section 4.

Fig 1: Sketched $F$-test for group testing. Here $n = 200, p = 500$ and $q = 50$.

**Distribution assumptions.** In our settings, each row of the design matrix $X$ and noise vector $Z$ follow Gaussian distributions and are independent of each other. We shall make these more explicit in our theorem statements. We will also make it clear that Theorem 1 and 3 hold beyond Gaussian settings, and a set of weaker distributional assumptions are stated in Appendix A. Moreover, we remark that the *Bayesian-type assumption (A)* in Proposition 1 is made only for illustrative purposes and is not used outside of line 192-215. Sorry for the confusion. In fact, the specific assumption (A) is invoked to provide intuition for the ARE expression in (8). We will also make it clear that we apply the *frequentist* approach throughout the entire paper, and all the theorems hold for any specific $(\beta, \Sigma, \sigma)$ under mild conditions.

**More discussions of our simulations.** In most of our simulations, we consider cases where the eigenvalues of $\Sigma$ enjoy a decaying structure, in which setting, the designs are of intrinsically lower dimensions. For such decaying structures, the U-statistics type tests (e.g. CGZ, ZC) by design are not suitable, therefore are not as competitive. These simulation results support our theoretical findings in Theorem 2 and 3. We emphasize that the optimality of our procedure relies on the intrinsic low-dimensional structure (in Definition 2/Appendix B); when there is no such structure, it is impossible to do feature-dimension reduction without losing information, and the optimal test for the global null remains open.

**Novelty of the method and theory.** As far as we know, the proposed procedure is the first attempt to analyze in details how sketching techniques work for testing regression coefficients. By a novel definition of the intrinsic dimension, we provide a systematic approach to determine the optimal sketching dimensions. While our theoretical results are built upon random matrix theory and the minimax decision framework, some technical lemmas introduced new techniques and are of independent interests.

[Meta-Review · NeurIPS 2020]

Three out of four reviewers are positive about the paper. I am suggesting acceptance. That said, reviewer R2 is a leading expert on the randomization inference and I strongly suggest that you address their comments in the final version of the manuscript. In particular, the experiments are rather limiting and should be expanded on.